



# Learning from satellite observations: increased understanding of catchment processes through stepwise model improvement

Petra Hulsman[1], Hubert H.G. Savenije[1], Markus Hrachowitz[1]

[1]Water Resources Section, Faculty of Civil Engineering and Geosciences, Delft University of Technology, Stevinweg 1, 2628 CN Delft, The Netherlands

*Correspondence to*: Petra Hulsman (p.hulsman@tudelft.nl)

**Abstract.** Satellite observations can provide valuable information for a better understanding of hydrological processes and thus serve as valuable tools for model structure development and improvement. While model calibration and evaluation has in recent years started to make increasing use of spatial, mostly remotely-sensed information, model structural development largely remains to rely on discharge observations at basin outlets only. Due to the ill-posed inverse nature and the related equifinality issues in the modelling process, this frequently results in poor representations of the spatiotemporal heterogeneity of system-internal processes, in particular for large river basins. The objective of this study is thus to explore the value of remotely-sensed, gridded data to improve our understanding of the processes underlying this heterogeneity and, as a consequence, their quantitative representation in models through a stepwise adaptation of model structures and parameters. For this purpose, a distributed, process-based hydrological model was developed for the study region, the poorly gauged Luangwa river basin. As a first step, this benchmark model was calibrated to discharge data only and, in a post-calibration evaluation procedure, tested for its ability to simultaneously reproduce (1) the basin-average temporal dynamics of remotely-sensed evaporation and total water storage anomalies, and (2) their temporally-averaged spatial pattern. This allowed the diagnosis of model structural deficiencies in reproducing these temporal dynamics and spatial patterns. Subsequently, the model structure was adapted in a step-wise procedure, testing five additional alternative process hypotheses that could potentially better describe the observed dynamics and pattern. These included, on the one hand, the addition and testing of alternative formulations of groundwater upwelling into wetlands as function of the water storage and, on the other hand, alternative spatial discretizations of the groundwater reservoir. Similar to the benchmark, each alternative model hypothesis was, in a next step, calibrated to discharge only and tested against its ability to reproduce the observed spatiotemporal pattern in evaporation and water storage anomalies. In a final step, all models were re-calibrated to discharge, evaporation and water storage anomalies simultaneously. The results indicated that (1) the benchmark model (Model A) could reasonably well reproduce the time series of observed discharge, basin-average evaporation and total water storage. In contrast, it poorly represented time series of evaporation in wetland dominated areas as well as the spatial pattern of evaporation and total water storage. (2) Step-wise adjusting the model structure (Models B – F) suggested that Model F, allowing for upwelling groundwater from a distributed representation of the groundwater reservoir and (3) simultaneously calibrating the model with respect to multiple variables, i.e. discharge, evaporation and total water storage anomalies, provided the best representation of all these variables with respect to their temporal dynamics and spatial pattern, except for the basin-average temporal dynamics in the total water storage anomalies. It was shown that satellite-based evaporation and total water storage anomaly data are not only valuable for multi-criteria calibration, but can play an important role in improving our understanding of hydrological processes through diagnosing model deficiencies and step-wise model structural improvement.





## 1. Introduction

Traditionally, discharge observations at basin outlets are used for hydrological model development and calibration, which

can be a robust strategy in small watersheds with relatively uniform characteristics such as topography and land cover, but not for larger, heterogeneous basins (Daggupati et al., 2015; Blöschl and Sivapalan, 1995). As a result, temporal dynamics of discharge may be well reproduced. This however, does not ensure that the spatial pattern and temporal dynamics of model internal storage and flux variables provide a meaningful representation of their real pattern and dynamics (Garavaglia et al., 2017; Hrachowitz et al., 2014; Beven, 2006; Kirchner, 2006; Gupta et al., 2008; Clark et al., 2008). Especially in large,

poorly gauged basins this traditional model calibration and testing method is likely to result in a poor representation of spatial variability (Daggupati et al., 2015) due to equifinality and the related the boundary flux problem (Beven, 2006).

An increasing number of satellite-based observations have become available over the last decade, giving us insight into a wide range of hydrology-relevant variables, including precipitation, total water storage anomalies, evaporation, surface soil moisture or river width (Xu et al., 2014; Jiang and Wang, 2019). These data are increasingly used as model forcing or for

parameter selection and model calibration (e.g. Mazzoleni et al., 2019; Li et al., 2015; Tang et al., 2019).

Many studies used a single satellite product in the calibration procedure, some of them additionally using discharge data, others not. For instance, hydrological models have been calibrated with respect to evaporation (e.g. Vervoort et al., 2014; Winsemius et al., 2008; Immerzeel and Droogers, 2008; Odusanya et al., 2019; Bouaziz et al., 2018), water storage anomalies from GRACE (Gravity Recovery and Climate Experiment, Werth et al., 2009), river width (Sun et al., 2018;

Revilla-Romero et al., 2015) or river altimetry (Sun et al., 2015; Getirana, 2010; Michailovsky et al., 2013; Hulsman et al., 2019).

Other studies simultaneously calibrated hydrological models with respect to multiple remotely-sensed variables, but only exploiting basin-average time series, without consideration for spatial pattern (e.g. López et al., 2017; Nijzink et al., 2018; Kittel et al., 2018; Milzow et al., 2011). On the other hand, some studies exclusively calibrated models to spatial pattern of

the observed variables (Demirel et al., 2018; Mendiguren et al., 2017; Stisen et al., 2011; Koch et al., 2016; Zink et al., 2018). As most satellite-based observations such as evaporation are not measured directly but are themselves a result of underlying models using satellite data as input (Xu et al., 2014), more focus has been recently placed on calibration to the relative spatial variability instead of using absolute magnitudes (Stisen et al., 2011; van Dijk and Renzullo, 2011; Dembélé et al., 2020).

To fully exploit the information content of satellite-based observations, simultaneous model calibration on both, temporal dynamics and spatial pattern of multiple variables has the potential to improve the representation of spatiotemporal variability and, linked to that, their underlying model internal processes and therefore the model realism (Rakovec et al., 2016; Herman et al., 2018; Rientjes et al., 2013). Strikingly, only a few studies so far used satellite-based observations to calibrate with respect to the temporal and spatial variation simultaneously (Dembélé et al., 2020; Rajib et al., 2018).

In general, most studies that made use of remotely-sensed data for model applications have exclusively addressed the problem of parameter selection and thus model calibration. However, as models are always abstract and simplified representations of reality, every model structure needs to be understood as a hypothesis to be tested (Clark et al., 2011; Fenicia et al., 2011; Hrachowitz and Clark, 2017). Yet, most studies on model structural improvement have so far only relied on spatially aggregated variables (Hrachowitz et al., 2014; Nijzink et al., 2016; Fenicia et al., 2008; Kavetski and Fenicia,

2011), while spatial data remain rarely used for that purpose (e.g. Roy et al., 2017; Fenicia et al., 2016).

The overall objective of this paper is therefore to explore the simultaneous use of spatial pattern and temporal dynamics of satellite-based evaporation and total water storage observations for a step-wise structural improvement and calibration of hydrological models for a large river systems in a semi-arid, data scarce region. More specifically, we tested the research hypotheses that (1) spatial pattern and temporal dynamics in satellite-based evaporation and water storage anomaly data

contain relevant information to diagnose and to iteratively improve on model structural deficiencies and that (2) these data,



when simultaneously used with discharge data for calibration, do contain sufficient information for a more robust parameter selection.

## 2. Site description

The Luangwa River in Zambia is a large, mostly unregulated tributary of the Zambezi with a length of about 770 km (Figure
1). This poorly gauged river basin has an area of 159,000 km$^2$ which is mostly covered with deciduous forest, shrubs and savanna and where an elevation difference up to 1850 m can be found between the highlands and low lands along the river (The World Bank, 2010; Hulsman et al., 2019). In this semi-arid basin, the mean annual evaporation (1555 mm yr$^{-1}$) exceeds the mean annual precipitation (970 mm yr$^{-1}$).

The Luangwa River flows into the Zambezi upstream of the Cahora Bassa Dam which is used for hydropower production,
and flood and drought protection. The operation of this dam is very difficult since there is only limited information available from the poorly gauged upstream tributaries (SADC, 2008; Schleiss and Matos, 2016). As a result, the local population has in the past suffered from severe floods and droughts (ZAMCOM et al., 2015; Schumann et al., 2016).

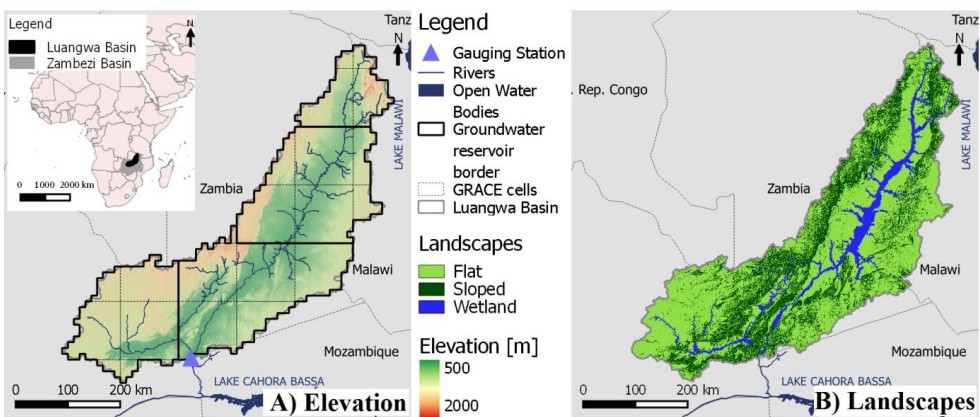

**Figure 1: Map of the Luangwa River Basin in Zambia with A) the elevation, groundwater reservoir units at 0.1° resolution and 1°**
**grid according to GRACE, and B) the main landscape types**

### 2.1 Data availability

#### 2.1.1 In-situ discharge observations

Historical daily in-situ discharge data was available from the Zambian Water Resources Management Authority at the Great East Road Bridge gauging station, located at 30° 13' E and 14° 58' S (Figure 1), for the time period 2004 to 2016 yet
containing considerable gaps resulting in a temporal coverage of 31%.

#### 2.1.2 Spatially gridded observation

Spatially gridded data were used for a topography-based landscape classification into hydrological response units (HRU, Savenije, 2010), as model forcing (precipitation and temperature) and for parameter selection (evaporation and total water storage, see Table 1).

More specifically, topography was extracted from GMTED with a spatial resolution of 0.002° (Danielson and Gesch, 2011). Daily precipitation data was extracted from CHIRPS (Climate Hazards Group InfraRed Precipitation with Station) with a spatial resolution of 0.05°. Monthly temperature data extracted from CRU at a spatial resolution of 1° was used to estimate the potential evaporation applying the Hargreaves method (Hargreaves and Samani, 1985; Hargreaves and Allen, 2003). These monthly observations were interpolated to daily timescale using daily averaged in-situ temperature measured at two





locations with the coordinates 28° 30' E, 14° 24' S and 32° 35' E, 13° 33' S. The satellite-based total evaporation data was
extracted from WaPOR (Water Productivity Open-access portal, FAO, 2018) version 1.1 as it proved to perform well in
African river basins (Weerasinghe et al., 2019). This product was available on 10-day temporal and 250 m spatial resolution.
Satellite-based observations on the total water storage anomalies were extracted from the Gravity Recovery and Climate
Experiment (GRACE). With two identical GRACE satellites, the variations in the Earth's gravity field were measured to
detect regional mass changes which are dominated by variations in the terrestrial water storage after having accounted for
atmospheric and oceanic effects (Landerer and Swenson, 2012; Swenson, 2012). In this study, the long term bias between
the discharge, evaporation (WaPOR) and total water storage anomalies (GRACE) was corrected by multiplying the
evaporation with a correction factor of 1.08 to close the long term water balance.

The gridded information provided for the precipitation, temperature and evaporation were rescaled to the model resolution of
0.1°. If the resolution of the satellite product was higher than 0.1°, then the mean of all cells located within each model cell
was used. Otherwise, each cell of the satellite product was divided into multiple cells such that the model resolution is
obtained, retaining the original value. In contrast, the modelled total water storage was rescaled to 1°, the resolution of the
GRACE data set, by taking the mean.

**Table 1: Data used in this study**

|  | Time period | Time Resolution | Spatial resolution | Product Name | Source |
|---|---|---|---|---|---|
| **Digital elevation map** | NA | NA | 0.002° | GMTED | (Danielson and Gesch, 2011) |
| **Precipitation** | 2002 – 2016 | Daily | 0.05° | CHIRPS | (Funk et al., 2014) |
| **Temperature** | 2002 – 2016 | Monthly | 0.5° | CRU | (University of East Anglia Climatic Research Unit et al., 2017) |
| **Evaporation** | 2009 – 2016 | 10-day | 0.00223° | WaPOR | (FAO and IHE Delft, 2019; FAO, 2018) |
| **Total water storage** | 2002 – 2016 | Monthly | 1° | GRACE | (Swenson, 2012; Swenson and Wahr, 2006; Landerer and Swenson, 2012) |
| **Discharge (Luangwa Bridge gauging station)** | 2004 – 2016 | Daily | NA | NA | WARMA |

**3. Modelling approach**

A previously developed and tested (Hulsman et al., 2019) distributed, process-based hydrological model was implemented
for the Luangwa Basin, see Section 3.1 for more information. This benchmark model (Model A) was calibrated with respect
to discharge for the time period 2002 – 2012 and validated for the time period 2012 – 2016 with respect to discharge,
evaporation and total water storage anomalies. Then, the model was calibrated with respect to all above variables, hence
discharge, evaporation and total water storage anomalies simultaneously, for the time period 2002 – 2012 and validated with
respect to the same variables for the time period 2012 – 2016. Model deficiencies were then diagnosed for this benchmark
model (Model A) based on the results of both calibration strategies.

Next, model structure changes were applied creating Models B – D to improve the deficiencies found in Model A. These
changes concerned the groundwater upwelling into the unsaturated zone as explained in Section 4.2.1. The same calibration
and validation strategies as applied to Model A were applied to Models B – D. Model improvements were evaluated and
further deficiencies were diagnosed for these models based on the calibration and validation results.

To improve the deficiencies diagnosed in Models B – D, further model structural changes, i.e. increased levels of spatial
discretisation of the saturated zone as explained in Section 4.3.1, resulted in Models E and F. Similar to the previous models,





the same calibration and validation strategies were applied to Models E and F, and model improvements and deficiencies
were diagnosed based on the calibration and validation model performances.

The calculation of the model performance with respect to discharge, evaporation and total water storage are explained in
Section 3.2. The calibration and validation procedures are described in Sections 3.3 and 3.4.

### 3.1 Hydrological models

### 3.1.1 Benchmark model (Model A)

This model is a process-based hydrological model developed in a previous study by Hulsman et al. (2019) for the Luangwa
basin. In this model, the water accounting was distributed by discretizing the basin and using spatially distributed forcing
data while the same model structure and parameter set were used for the entire basin. Each $0.1^o$ x $0.1^o$ model cell was then
further discretized into functionally distinct landscape classes, i.e. hydrological response units (HRU), inferred from
topography (Figure 1B), but connected by a common groundwater component (Euser et al., 2015) following the FLEX-Topo
modelling concept (Savenije, 2010) which was previously successfully applied in many different and climatically contrasting
regions (Gao et al., 2014; 2016; Gharari et al., 2014; Nijzink et al., 2016). Here, the landscape was classified based on the
local slope and "Height-above-the-nearest-drainage" (HAND, Rennó et al., 2008) into sloped areas (slope $\geq$ 4%), flat areas
(slope < 4%, HAND $\geq$ 11 m) and wetlands (slope < 4%, HAND < 11 m). For this purpose, the drainage network was derived
from a digital elevation map extracted from GMTED (Section 2.1.2) using a flow accumulation map after having burned-in a
river network map extracted from OpenStreetMap (https://wiki.openstreetmap.org/wiki/Shapefiles) to obtain an as accurate
as possible drainage network as done successfully in previous studies (Seyler et al., 2009). According to this classification,
the wetland areas covered 8% of the basin, flat areas 64% and sloped areas 28% (Figure 1).

The model consisted of different storage components schematised as reservoirs representing interception and unsaturated
storage, as well as a slow responding reservoir, representing the groundwater and a fast responding reservoir (Figure 2). The
water balance for each reservoir and the associated constitutive equations are summarised in Table 2. The individual model
structures of each parallel HRU were very similar. Functional differences between HRUs were thus mostly accounted for by
different parameter sets. To allow the use of partly overlapping prior parameter distributions while maintaining relationships
between parameters of individual HRUs that are consistent with our physical understanding of the system and to limit
equifinality, model process constraints (Gharari et al., 2014; Hrachowitz et al., 2014) were applied for several parameters
(Table 3). For instance, in the Luangwa Basin, the sloped areas are dominated by dense vegetation, suggesting higher
interception capacities and larger storage capacities in the unsaturated zone compared to the remaining part of the basin. In
addition, for each HRU the model structure was adjusted where necessary to include processes unique to that area. For
instance, water percolates and recharges the groundwater system in sloped and flat areas whereas in wetlands this was
assumed to be negligible due to groundwater tables that are very shallow and thus close to the surface.

The runoff was first calculated for each individual grid cell. A simple routing scheme based on the flow direction and
constant flow velocity as calibration parameter was applied to estimate the flow at the outlet. In total, this model consisted of
16 calibration parameters with uniform prior distributions and constraints as summarized in Table 3.

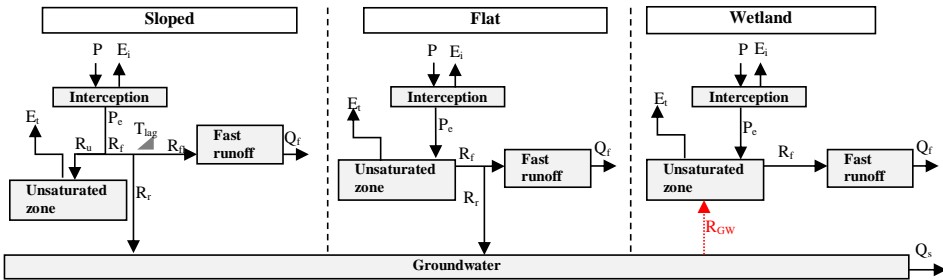

**Figure 2: Schematisation of the model structure applied to each grid cell. Symbol explanation: precipitation ($P$), effective precipitation ($P_e$), interception evaporation ($E_i$), plant transpiration ($E_t$), infiltration into the unsaturated root zone ($R_u$), drainage to fast runoff component ($R_f$), delayed fast runoff ($R_{fl}$), lag time ($T_{lag}$), groundwater recharge ($R_r$), upwelling groundwater flux ($R_{GW}$), fast runoff ($Q_f$), groundwater/slow runoff ($Q_s$).**


**Table 2: Equations applied in the hydrological model. Fluxes [mm d$^{-1}$]: precipitation ($P$), effective precipitation ($P_e$), potential evaporation ($E_p$), interception evaporation ($E_i$), plant transpiration ($E_t$), infiltration into the unsaturated zone ($R_u$), drainage to fast runoff component ($R_f$), delayed fast runoff ($R_{fl}$), groundwater recharge ($R_r$ for each relevant HRU and $R_{r,tot}$ combining all relevant HRUs), groundwater upwelling ($R_{GW}$ for each relevant HRU and $R_{GW,tot}$ combining all relevant HRUs), fast runoff ($Q_f$ for each**
**HRU and $Q_{f,tot}$ combining all HRUs), total runoff ($Q_m$). Storages [mm]: storage in interception reservoir ($S_i$), storage in unsaturated root zone ($S_u$), storage in groundwater/slow reservoir ($S_s$), storage in fast reservoir ($S_f$). Parameters: interception capacity ($I_{max}$) [mm], maximum upwelling groundwater ($C_{max}$) [mm d$^{-1}$], maximum root zone storage capacity ($S_{umax}$) [mm], reference storage in the saturated zone ($S_{s,ref}$) [mm], splitter ($W$) [-], shape parameter ($\beta$) [-], transpiration coefficient ($C_e$) [-], time lag ($T_{lag}$) [d], exponent ($\gamma$) [-], reservoir time scales [d] of fast ($K_f$) and slow ($K_s$) reservoirs, areal weights for**
**each grid cell ($p_{HRU}$) [-], time step ($\Delta t$) [d]. Model calibration parameters are shown in bold letters in the table below. The equations were applied to each hydrological response unit (HRU) unless indicated differently.**

| Reservoir system | Water balance equation | Equation | Process functions | Equation |
|---|---|---|---|---|
| **Interception** | $\frac{\Delta S_i}{\Delta t} = P - P_e - E_i$ | (1) | $E_i = \min\left(E_p, \min\left(P, \frac{I_{max}}{\Delta t}\right)\right)$ | (2) |
| | | | $P_e = P - E_i$ | (3) |
| **Unsaturated zone** | Sloped:<br>$\frac{\Delta S_u}{\Delta t} = R_u - E_t$ | (4) | $E_t = \min\left((E_p - E_i), \min\left(\frac{S_u}{\Delta t}, (E_p - E_i) \cdot \frac{S_u}{S_{u,max}} \cdot \frac{1}{C_e}\right)\right)$ | (5) |
| | Flat:<br>$\frac{\Delta S_u}{\Delta t} = P_e - E_t - R_f$ | (6) | Model A: $R_{GW} = 0$ | (7) |
| | Wetland:<br>$\frac{\Delta S_u}{\Delta t} = P_e - E_t - R_f + R_{GW}$ | (8) | Model B: $R_{GW} = \min\left(\left(1 - \frac{S_u}{S_{u,max}}\right) \cdot C_{max}, \frac{\frac{S_s}{\Delta t}}{p_{HRU}}\right)$ | (9) |
| | | | Model C, E, F: $R_{GW} = \min\left(\frac{\min(S_s, S_{s,ref})}{S_{s,ref}} \cdot C_{max}, \frac{\frac{S_s}{\Delta t}}{p_{HRU}}\right)$ | (10) |
| | | | Model D: $R_{GW} = \min\left(\left(\frac{\min(S_s, S_{s,ref})}{S_{s,ref}}\right)^\gamma \cdot C_{max}, \frac{\frac{S_s}{\Delta t}}{p_{HRU}}\right)$ | (11) |
| | | | if $S_u + R_{GW} \cdot \Delta t > S_{u,max}$: $R_{GW} = \frac{S_{u,max} - S_u}{\Delta t}$ | (12) |
| | | | Hillslope:<br>$R_u = (1 - C) \cdot P_e$ | (13) |
| | | | $C = 1 - \left(1 - \frac{S_u}{S_{u,max}}\right)^\beta$ | (14) |
| **Fast runoff** | $\frac{\Delta S_f}{\Delta t} = R_{fl} - Q_f$ | (15) | $Q_f = \frac{S_f}{K_f}$ | (16) |
| | | | Flat/Wetland:<br>$R_f = \frac{\max(0, S_u - S_{u,max})}{\Delta t}$ | (17) |
| | | | $R_{fl} = R_f$ | (18) |
| | | | Sloped:<br>$R_f = (1 - W) \cdot C \cdot P_e$ | (19) |
| | | | $R_{fl} = R_f * f(T_{lag})$ | (20) |
| **Groundwater** | $\frac{\Delta S_s}{\Delta t} = R_{r_{tot}} - R_{GW_{tot}} - Q_s$ | (21) | $R_r = W \cdot C \cdot P_e$ | (22) |
| | | | $R_{r_{tot}} = \sum_{HRU} p_{HRU} \cdot R_r$ | (23) |
| | | | $R_{GW_{tot}} = \sum_{HRU} p_{HRU} \cdot R_{GW}$ | (24) |
| | | | $Q_s = \frac{S_s}{K_s}$ | (25) |
| **Total runoff** | $Q_m = Q_s + Q_{f_{tot}}$ | (26) | $Q_{f_{tot}} = \sum_{HRU} p_{HRU} \cdot Q_f$ | (27) |
| **Supporting literature** | (Hulsman et al., 2019; Gharari et al., 2014; Gao et al., 2014; Euser et al., 2015) | | | |





**Table 3: Model parameter and ranges (Hulsman et al., 2019)**

| Landscape class | Parameter | min | max | Unit | Constraint | Comment |
|---|---|---|---|---|---|---|
| **Entire basin** | $C_e$ | 0 | 1 | - | | |
| | $K_s$ | 50 | 200 | d | | |
| | $S_{sref}$ | 100 | 500 | mm | | Only for Models C to F |
| **Flat** | $I_{max}$ | 0 | 5 | mm d$^{-1}$ | | |
| | $S_{u,max}$ | 300 | 1000 | mm | | |
| | $K_f$ | 10 | 12 | d | | |
| | $W$ | 0.5 | 0.95 | - | | |
| **Sloped** | $I_{max}$ | 0 | 5 | mm d$^{-1}$ | $I_{max,sloped} > I_{max,flat}$ | |
| | $S_{umax}$ | 300 | 1000 | mm | $S_{umax,sloped} > S_{umax,flat}$ | |
| | $\beta$ | 0 | 2 | - | | |
| | $T_{lag}$ | 1 | 5 | d | | |
| | $K_f$ | 10 | 12 | d | | |
| | $W$ | 0.5 | 0.95 | - | $W_{sloped} > W_{flat}$ | |
| **Wetland** | $I_{max}$ | 0 | 5 | mm d$^{-1}$ | $I_{max,wetland} < I_{max,sloped}$ | |
| | $S_{umax}$ | 10 | 500 | mm | $S_{umax,wetland} < S_{umax,sloped}$ | |
| | $K_f$ | 10 | 12 | d | | |
| | $C_{max}$ | 0.1 | 5 | mm d$^{-1}$ | | Only for Models B to F |
| | $\gamma$ | 0.01 | 0.5 | - | | Only for Model D |
| **River profile** | $v$ | 0.01 | 5.0 | m s$^{-1}$ | | |

### 3.1.2 First model adaptation (Models B – D)

As first model adaption, groundwater upwelling ($R_{GW}$) was added in wetland areas (see Figure 2). This upwelling groundwater was made (1) a linear function of the water content in the unsaturated reservoir (Model B, Eq.9 in Table 2), (2) a linear function of the water content in the slow responding reservoir (Model C, Eq.10) and (3) a non-linear function of the water content in the slow responding reservoir (Model D, Eq.11). As a result, upwelling water from the saturated zone feeds the unsaturated zone, controlled by the water content in the unsaturated (Model B) or in the saturated zone (Models C and D), and thus increasing the water availability for transpiration from the unsaturated zone in wetland areas. Compared to the benchmark Model A, Model B introduces one additional calibration parameter, Model C two and Model D three (Tables 2 and 3). See Section 4.2 for more detailed information on the reasons for and processes behind these model adjustments.

### 3.1.3 Second model adaptation (Models E – F)

As second model adaptation, the spatial resolution of the slow responding reservoir was gradually increased from lumped (Models A – D) to semi-distributed (Model E) and fully distributed (Model F). In Model E, the slow responding reservoir was divided into four units as visualised in Figure 1A, whereas in Model F it was further discretized into a grid of 10 x 10 km$^2$, equivalent to the remaining parts of the model. For both alternative formulations, Models E and F respectively, the slow reservoir timescales $K_s$ remained constant throughout the basin to limit the number of calibration parameters. For both Models E and F, groundwater upwelling was included according to Eq.10 (Table 2), hence using Model C as basis, introducing two additional calibration parameters compared to the benchmark Model A (Tables 2 and 3). See Section 4.3 for more detailed information on the reasons for and processes behind these model adjustments.

### 3.2 Model performance metrics

#### 3.2.1 Discharge

The model performance with respect to discharge was evaluated using eight distinct signatures simultaneously characterizing the observed discharge data (Hulsman et al., 2019; Euser et al., 2013). The model performance measure was based either on the Nash-Sutcliffe efficiency ($E_{NS,\theta}$, Eq.28 in Table 4) or the relative error ($E_{R,\theta}$, Eq.29) depending on the individual signature. The resulting performance metrics for the eight signatures then included the Nash-Sutcliffe efficiencies of the daily discharge time series ($E_{NS,Q}$), its logarithm ($E_{NS,logQ}$), the flow duration curve ($E_{NS,FDC}$), its logarithm ($E_{NS,logFDC}$) and of the autocorrelation function of daily flows ($E_{NS,AC}$) and the relative errors of the mean seasonal runoff coefficient during dry and wet periods ($E_{R,RCdry}$, $E_{R,RCwet}$) and of the rising limb density of the hydrograph ($E_{R,RLD}$). All these signatures were





combined into an overall performance metric based on the Euclidian distance to the "perfect" model ($D_{E,Qcal}$, Eq.31). In absence of more information and to obtain balanced solutions, all individual performance metrics were equally weighted in Eq.31. Here, a $D_{E,Qcal} = 1$ indicates a perfect fit.

The discharge data availability was very limited during the validation time period (2012 – 2016). As a result, hydrological years were not fully captured resulting in incomplete information on the hydrologic signatures such as rising limb density or auto correlation function. That is why the overall model performance ($D_{E,Qval}$) was calculated using the signatures $E_{NS,Q}$, $E_{NS,logQ}$, $E_{NS,FDC}$ and $E_{NS,logFDC}$ excluding $E_{R,RCdry}$, $E_{R,RCwet}$, $E_{R,RLD}$ and $E_{NS,AC}$. It is therefore important to note that $D_{E,Qcal}$ cannot be meaningfully compared with $D_{E,Qval}$. Instead, following the overall objective of the analysis, $D_{E,Qval}$ of the different

alternative model hypothesis were compared to evaluate the differences between the models.

**Table 4: Overview of equations used to calculate model performance**

| Name | Objective Function | Equation | Variable explanation |
|---|---|---|---|
| Nash-Sutcliffe efficiency | $E_{NS,\theta} = 1 - \frac{\sum_t (\theta_{mod}(t) - \theta_{obs}(t))^2}{\sum_t (\theta_{obs}(t) - \overline{\theta_{obs}})^2}$ | (28) | $\theta$ variable |
| Relative error | $E_{R,\theta} = 1 - \frac{|\theta_{mod} - \theta_{obs}|}{\theta_{obs}}$ | (29) | |
| Spatial efficiency metric | $E_{SP} = \frac{1}{t_{max}} * \sum_t 1 - \sqrt{(\alpha - 1)^2 + (\beta - 1)^2 + (\gamma - 1)^2}$ With: $\alpha = \rho(\varphi_{obs}, \varphi_{mod})$ $\beta = \frac{\sigma_{obs}/\mu_{obs}}{\sigma_{mod}/\mu_{mod}}$ $\gamma = \left(\sum_{i=0}^{i=n} \min(K_i, L_i)\right) * \left(\sum_{i=0}^{i=n} K_i\right)^{-1}$ | (30) | $\alpha$ Pearson correlation coefficient $\varphi_{obs}$, $\varphi_{mod}$ observed/modelled map $\beta$ coefficient of variation $\sigma$ standard deviation $\mu$ mean $\gamma$ fraction of histogram intersection between $K$ and $L$ K observed histogram L modelled histogram $n = 100$ bins $t$ time step within the dry season with maximum $t_{max}$ |
| Euclidian distance over multiple signatures | $D_{E,Q} = 1 - \sqrt{\frac{1}{(N+M)}\left(\sum_n (1 - E_{NS,\theta_n})^2 + \sum_m (1 - E_{R,\theta_m})^2\right)}$ | (31) | $n$ signatures evaluated with Eq.28 with maximum $N$ $m$ signatures evaluated with Eq.29 with maximum $M$ |
| Euclidian distance over multiple variables | $D_{E,ESQ} = 1 - \sqrt{\frac{1}{N}(\sum_n (1 - E_n)^2)}$ | (32) | $n$ variables maximum $N$ $E_n$ model performance metric of variable $n$ |

### 3.2.2 Evaporation and total water storage

The model performance was also evaluated with respect to both the temporal dynamics and the spatial pattern of evaporation and total storage, respectively. For this purpose, satellite-based evaporation data (WaPOR) was used on 10-day time scale, and total water storage anomaly data (GRACE) on monthly time scale.

### Temporal variation

To quantify the models' skill to reproduce the temporal dynamics of evaporation and total water storage anomalies, the respective Nash-Sutcliffe efficiencies (Eq. 28) were used as performance metrics. This performance metric was applied to assess the models' skill to reproduce the basin-average time series of evaporation and total water storage anomalies, i.e.





$E_{\text{NS,Basin,E}}$ and $E_{\text{NS,Basin,S}}$, respectively. Similarly, the models' performance to mimic the dynamics of evaporation in all grid cells dominated by the wetland HRU was calculated with the Nash-Sutcliffe efficiency ($E_{\text{NS,Wetland,E}}$). Grid cells were considered as wetland dominated if they were completely covered by wetlands, hence if $p_{\text{HRU}} = 1$ with $p_{\text{HRU}}$ the areal weight of wetland areas within that cell. With respect to evaporation, the flux was normalised first with Eq.33 to emphasize temporal variations rather than absolute values in an attempt to reduce bias related errors in the observation:

$$E_{\text{normalised}} = \frac{E - E_{\min}}{E_{\max} - E_{\min}} \tag{33}$$

**Spatial variation**

The model performance with respect to the spatial pattern of evaporation and total water storage anomalies was calculated with the spatial efficiency metrics $E_{\text{SP,E}}$ and $E_{\text{SP,S}}$ (Eq.30), respectively, which was successfully used in previous studies (Demirel et al., 2018; Koch et al., 2018). The spatial model performance was first calculated for each time step within the dry period which was in September/October and then averaged to obtain the overall model performance ($E_{\text{SP}}$, Eq.30). The spatial pattern was averaged over the dry season to minimize the effect of precipitation errors.

### 3.2.3 Multi-variable

The overall potential of the models to simultaneously reproduce the temporal dynamics as well as the spatial pattern of all observed variables, i.e. discharge, evaporation and total water storage anomalies, was tested with the overall model performance metric $D_{\text{E,ESQ}}$. This metric was the Euclidian distance (Eq.32) of the following individual metrics: the temporal variation of the basin-average evaporation ($E_{\text{NS,Basin,E}}$) and total water storage anomalies ($E_{\text{NS,Basin,S}}$), spatial pattern of the evaporation ($E_{\text{SP,E}}$) and total water storage anomalies ($E_{\text{SP,S}}$) as well as discharge ($D_{\text{E,Q}}$). See Table 5 for an overview of all model performance metrics used in this study.

**Table 5: Overview of the applied model performance metrics**

| Data | Temporal dynamics/ Spatial pattern | Performance metric | Symbol and equation nr. | Calibration/validation |
|------|-----------------------------------|--------------------|-----------------------|------------------------|
| **Discharge** | Temporal dynamics | Euclidian distance over multiple signatures (combining $E_{\text{NS,Q}}$, $E_{\text{NS,logQ}}$, $E_{\text{NS,FDC}}$, $E_{\text{NS,logFDC}}$, $E_{\text{NS,AC}}$, $E_{\text{R,RCdry}}$, $E_{\text{R,RCwet}}$ and $E_{\text{R,RLD}}$) | $D_{\text{E,Qcal}}$ (Eq.31) | Calibration (2002 – 2012) |
| | Temporal dynamics | Euclidian distance over multiple signatures (combining $E_{\text{NS,Q}}$, $E_{\text{NS,logQ}}$, $E_{\text{NS,FDC}}$ and $E_{\text{NS,logFDC}}$) | $D_{\text{E,Qval}}$ (Eq.31) | Validation (2012 – 2016) |
| **Evaporation** | Temporal dynamics (basin-average) | Nash-Sutcliffe efficiency | $E_{\text{NS,Basin,E}}$ (Eq.28) | Validation (2012 – 2016) |
| | Temporal dynamics (wetland areas) | Nash-Sutcliffe efficiency | $E_{\text{NS,Wetland,E}}$ (Eq.28) | Validation (2012 – 2016) |
| | Spatial pattern | Spatial efficiency metric | $E_{\text{SP,E}}$ (Eq.30) | Validation (2012 – 2016) |
| **Total water storage anomalies** | Temporal dynamics (basin-average) | Nash-Sutcliffe efficiency | $E_{\text{NS,Basin,S}}$ (Eq.28) | Validation (2012 – 2016) |
| | Spatial pattern | Spatial efficiency metric | $E_{\text{SP,S}}$ (Eq.30) | Validation (2012 – 2016) |
| **Multi-variable (discharge, evaporation and total water storage anomalies)** | Combination | Euclidian distance over multiple variables (combining $D_{\text{E,Qcal}}$, $E_{\text{NS,Basin,E}}$, $E_{\text{SP,E}}$, $E_{\text{NS,Basin,S}}$ and $E_{\text{SP,S}}$) | $D_{\text{E,ESQcal}}$ (Eq.32) | Calibration (2002 – 2012) |
| | Combination | Euclidian distance over multiple variables (combining $D_{\text{E,Qval}}$, $E_{\text{NS,Basin,E}}$, $E_{\text{SP,E}}$, $E_{\text{NS,Basin,S}}$ and $E_{\text{SP,S}}$) | $D_{\text{E,ESQval}}$ (Eq.32) | Validation (2012 – 2016) |





### 3.3 Model calibration

In general, the model was calibrated by first running the model with $5 \cdot 10^5$ random parameter sets generated with a Monte-Carlo sampling strategy from uniform prior parameter distributions (Table 3). Then, the optimal and 5% best-performing parameter sets were selected according to the model performance metric as described in the previous section. The model was calibrated within the time period 2002 – 2012 with respect to 1) discharge ($D_{E,Qcal}$) and 2) all variables simultaneously ($D_{E,ESQcal}$).

### 3.4 Model validation

The model was validated with respect to discharge, evaporation and total water storage anomalies for the time period 2012 – 2016. During validation each variable was evaluated separately both temporally and spatially. This included the temporal variation of the basin-average evaporation ($E_{NS,Basin,E}$) and total water storage anomalies ($E_{NS,Basin,S}$), evaporation in wetland areas ($E_{NS,Wetland,E}$), spatial pattern of the evaporation ($E_{SP,E}$) and total water storage anomalies ($E_{SP,S}$) as well as discharge ($D_{E,Qval}$). In addition, the model was evaluated with respect to the overall performance ($D_{E,ESQval}$). This was done for the solutions from both calibration strategies.

## 4. Model results

### 4.1 Benchmark model (Model A)

#### 4.1.1 Discharge based calibration

For the benchmark model (Model A), the model performance of all model realizations following the first calibration strategy, i.e. calibrating to discharge, resulted in an optimum $D_{E,Qcal,opt} = 0.76$ and $D_{E,Qval} = 0.37$ during validation (Table 6, Figure 3). As shown in Figure 4, the main features of the hydrological response were captured reasonably well. However, particularly in the validation period, low flows were somewhat underestimated. Note that in 2013, the observed high flows were probably underestimated due to failures in the recording which resulted in a truncated top in the hydrograph and flat top in the flow duration curve during the validation time period (Figure 4) and which affect the validated model performance values ($D_{E,Qval}$). The range in the calibrated model performance with respect to each discharge signature separately is visualised in Figure S1 in the supplementary material.

The basin-average evaporation ($E_{NS, Basin,E} = 0.54$) and total water storage anomalies ($E_{NS, Basin,S} = 0.74$) were in general also reproduced rather well (Figures S3 and S5). In contrast, the model failed to mimic the evaporation dynamics in wetland dominated areas as it decreased rapidly to zero in the dry season in contrast to the observations ($E_{NS,Wetland,E} = 0.25$, Figure 5). Similarly, the spatial variability in evaporation ($E_{SP,E} = 0.17$) and water storage anomalies ($E_{SP,S} = -0.02$) were poorly captured as several areas were over- or underestimated (Figures 6 and 7). Note that in both figures the normalised evaporation and total water storage anomalies were plotted applying Eq.33 to emphasize relative spatial differences rather than absolute values.





**Table 6: Summary of model performance with respect to evaporation ($E_{NS,Basin,E}$, $E_{NS,Wetland,E}$ and $E_{SP,E}$), total water storage anomalies ($E_{NS,Basin,S}$ or $E_{SP,S}$), discharge ($D_{E,Qcal}$ and $D_{E,Qval}$) and all variables combined ($D_{E,ESQval}$): The parameter sets were selected based on discharge ($D_{E,Qcal}$).**

| | Calibration (2002 – 2012) | Validation (2012 – 2016) | | | | | | |
|---|---|---|---|---|---|---|---|---|
| | $D_{E,Qcal,opt}$ ($D_{E,Qcal,5/95\%}$) | $D_{E,Qval}$ ($D_{E,Qval,5/95\%}$) | $E_{NS,Basin,E}$ ($E_{NS,Basin,E,5/95}$) | $E_{NS,wetland,E}$ ($E_{NS,wetland,E,5/95}$) | $E_{SP,E}$ ($E_{SP,E,5/95}$) | $E_{NS,Basin,S}$ ($E_{NS,Basin,S,5/95}$) | $E_{SP,S}$ ($E_{SP,S,5/95}$) | $D_{E,ESQval}$ ($D_{E,ESQval,5/95}$) |
| **A** | 0.76 (0.54 – 0.68) | 0.37 (0.26 – 0.85) | 0.54 (0.34 – 0.57) | 0.25 (-0.14 – 0.58) | 0.17 (-0.37 – 0.04) | 0.74 (0.62 – 0.80) | -0.02 (-0.23 – 0.03) | 0.30 (0.12 – 0.29) |
| **B** | 0.75 (0.36 – 0.60) | 0.08 (-3.9 – 0.78) | 0.46 (0.34 – 0.63) | 0.29 (0.09 – 0.65) | 0.12 (-0.68 – -0.12) | 0.69 (0.61 – 0.82) | -0.07 (-0.20 – -0.08) | 0.21 (-1.3 – 0.27) |
| **C** | 0.79 (0.58 – 0.70) | 0.50 (0.27 – 0.85) | 0.50 (0.34 – 0.58) | 0.19 (-0.01 – 0.57) | 0.10 (-0.39 – -0.06) | 0.76 (0.62 – 0.81) | -0.08 (-0.23 – -0.04) | 0.32 (0.12 – 0.30) |
| **D** | 0.77 (0.53 – 0.68) | -1.7 (-2.4 – 0.84) | 0.36 (0.33 – 0.60) | 0.41 (0.11 – 0.62) | -0.04 (-0.57 – -0.10) | 0.63 (0.61 – 0.82) | -0.17 (-0.22 – -0.06) | -0.41 (-0.72 – -0.28) |
| **E** | 0.78 (0.58 – 0.70) | 0.81 (0.27 – 0.85) | 0.50 (0.34 – 0.58) | 0.07 (-0.04 – 0.59) | 0.05 (-0.39 – -0.05) | 0.77 (0.62 – 0.81) | -0.08 (-0.23 – -0.04) | 0.30 (0.12 – 0.29) |
| **F** | 0.91 (0.86 – 0.89) | 0.52 (0.12 – 0.74) | 0.61 (0.45 – 0.63) | 0.56 (-0.08 – 0.61) | -0.03 (-0.49 – -0.19) | 0.66 (0.44 – 0.71) | 0.08 (-0.07 – 0.13) | 0.31 (0.12 – 0.34) |

300

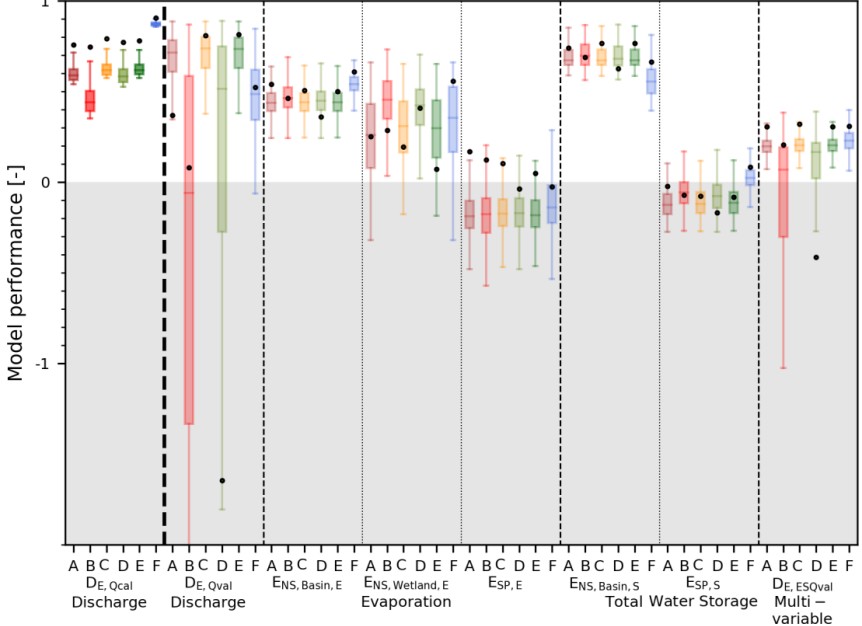

**Figure 3: Model performance with respect to discharge, evaporation and storage for all models. The model is calibrated to discharge (darker boxplots in the first column) and validated to the discharge, evaporation and storage (lighter boxplots). The dots represent the model performance using the "optimal" parameter set and the boxplot the range of the best 5% solutions both according to discharge ($D_{E,Qcal}$). The following performance metrics were used: 1) discharge using the overall model performance metric ($D_{E,Qcal}$ for calibration and $D_{E,Qval}$ for validation), 2) evaporation temporally basin-average ($E_{NS,Basin,E}$), 3) evaporation temporally wetland areas only ($E_{NS,Wetland,E}$), 4) evaporation spatially ($E_{SP,E}$), 5) storage temporally basin-average ($E_{NS,Basin,S}$), 6) storage temporally wetland areas only ($E_{NS,Wetland,S}$), 7) storage spatially ($E_{SP,S}$), and 8) the combination of evaporation, storage and discharge (combined metric $D_{E,ESQval}$).**

310

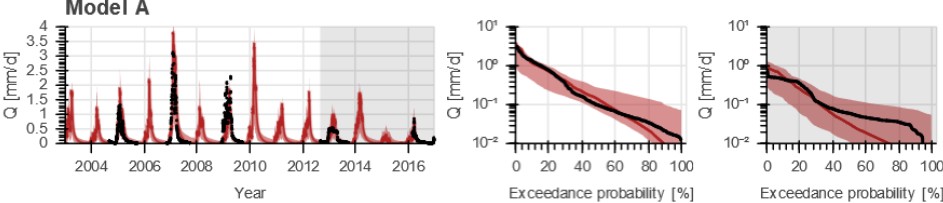




**Figure 4: Range of model solutions for Model A. The left panel shows the hydrograph and the right panel the flow duration curve of the recorded (black) and modelled discharge: the line indicates the solution with the highest calibration objective function with respect to discharge ($D_{E,Qcal}$) and the shaded area the envelope of the solutions retained as feasible. The data in the white area were used for calibration and the grey shaded area for validation.**

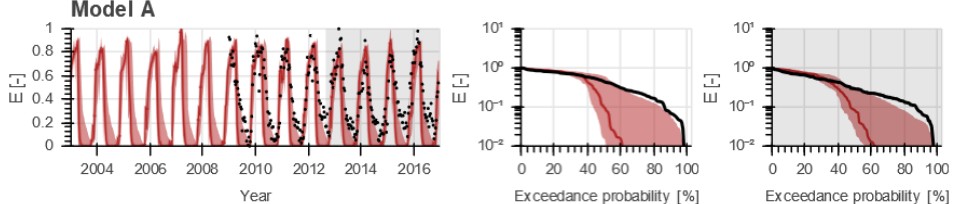

**Figure 5: Range of model solutions for Models A to F. The left panel shows the time series and the right panel the duration curve of the recorded (black) and modelled normalised evaporation for wetland dominated areas: the line indicates the solution with the highest calibration objective function with respect to discharge ($D_{E,Qcal}$) and the shaded area the envelope of the solutions retained as feasible. The data in the grey shaded area were used for validation.**

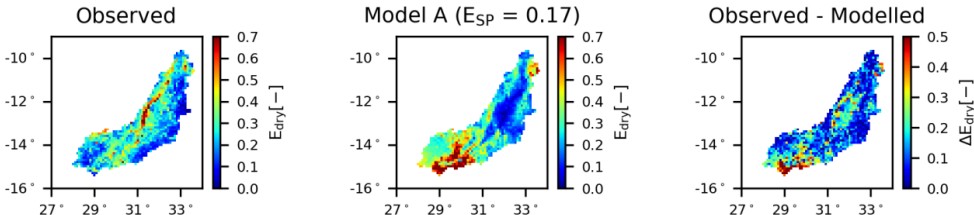

**Figure 6: Spatial variability of the normalised total evaporation for Model A averaged over all days within the dry season. The left panel shows the observation according to WaPOR data, the middle panel the model result using the "optimal" parameter set with respect to discharge ($D_{E,Qcal}$), and the right panel the difference between the observation and model.**

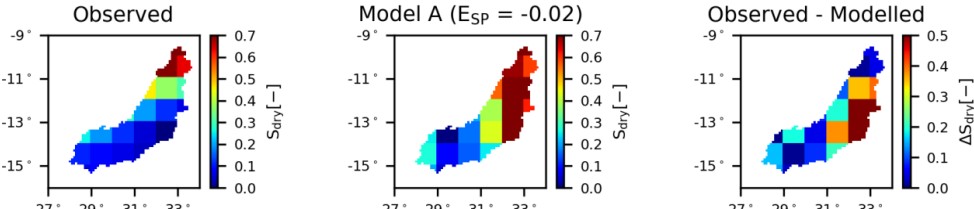

**Figure 7: Spatial variability of the normalised total water storage anomalies for Model A averaged over all days within the dry season. The left panel shows the observation according to GRACE data, the middle panel the model result using the "optimal" parameter set with respect to discharge ($D_{E,Qcal}$), and the right panel the difference between the observation and model.**

### 4.1.2 Multi-variable calibration

Calibrating with respect to multiple variables simultaneously in the second calibration strategy, resulted in a reduced model skill to simultaneously reproduce all flow signatures in the validation period with $D_{E,Qval} = 0.07$ (Table 7, Figures 8 and 9). Compared to the first calibration strategy, the simulated evaporation did not change significantly with respect to the temporal dynamics ($E_{NS,Wetland,E} = 0.27$, $E_{NS,Basin,E} = 0.57$) and spatial pattern ($E_{SP,E} = -0.18$). Evaporation from wetland dominated areas remained underestimated in the dry season (Figure 10) and large areas in the basin were still under- or overestimated (Figure 11). The reproduction of the total water storage anomalies decreased though, mostly with respect to the spatial pattern ($E_{SP,S} = -0.14$, Figure 12). On the other hand, when looking at the 5/95[th] percentile range instead of the "optimal" parameter set, then an improvement was observed in the spatial pattern in evaporation ($E_{SP,E,5/95} = -0.10 – 0.22$) and in total water storage ($E_{SP,S,5/95} = -0.17 – 0.08$, compare Tables 6 and 7).





**Table 7: Summary of the model performance with respect to evaporation ($E_{NS,Basin,E}$, $E_{NS,Wetland,E}$ and $E_{SP,E}$), total water storage anomalies ($E_{NS,Basin,S}$ or $E_{SP,S}$), discharge ($D_{E,Qval}$) and all variables combined ($D_{E,ESQval}$): Parameter sets selected based on multiple variables simultaneously ($D_{E,ESQcal}$).**

| | Calibration (2002 – 2012) | Validation (2012 – 2016) | | | | | | |
|---|---|---|---|---|---|---|---|---|
| | $D_{E,ESQcal,opt}$ ($D_{E,ESQcal,5/95}$) | $D_{E,Qval}$ ($D_{E,Qval,5/95\%}$) | $E_{NS,Basin,E}$ ($E_{NS,Basin,E,5/95}$) | $E_{NS,wetland,E}$ ($E_{NS,wetland,E,5/95}$) | $E_{SP,E}$ ($E_{SP,E,5/95}$) | $E_{NS,Basin,S}$ ($E_{NS,Basin,S,5/95}$) | $E_{SP,S}$ ($E_{SP,S,5/95}$) | $D_{E,ESQval}$ ($D_{E,ESQval,5/95}$) |
| **A** | 0.42 (0.28 − 0.36) | 0.07 (−1.4 − 0.80) | 0.57 (0.37 − 0.60) | 0.27 (−0.05 − 0.61) | 0.18 (−0.10 − 0.22) | 0.72 (0.60 − 0.77) | −0.14 (−0.17 − 0.08) | 0.21 (−0.25 − 0.32) |
| **B** | 0.40 (0.23 − 0.33) | 0.46 (−4.2 − 0.70) | 0.55 (0.39 − 0.63) | 0.56 (0.04 − 0.64) | 0.16 (−0.14 − 0.25) | 0.73 (0.61 − 0.79) | −0.16 (−0.17 − 0.09) | 0.28 (−1.4 − 0.29) |
| **C** | 0.44 (0.29 − 0.37) | 0.61 (−1.6 − 0.79) | 0.48 (0.37 − 0.61) | 0.51 (0.08 − 0.60) | 0.19 (−0.07 − 0.25) | 0.70 (0.60 − 0.77) | −0.03 (−0.16 − 0.09) | 0.33 (−0.31 − 0.33) |
| **D** | 0.43 (0.27 − 0.36) | −0.08 (−3.5 − 0.75) | 0.51 (0.38 − 0.62) | 0.59 (0.06 − 0.61) | 0.24 (−0.09 − 0.26) | 0.69 (0.60 − 0.78) | −0.04 (−0.16 − 0.09) | 0.21 (−1.1 − 0.32) |
| **E** | 0.43 (0.29 − 0.36) | 0.30 (−1.6 − 0.79) | 0.43 (0.38 − 0.61) | 0.30 (0.03 − 0.61) | 0.17 (−0.08 − 0.25) | 0.64 (0.60 − 0.77) | −0.02 (−0.16 − 0.10) | 0.27 (−0.31 − 0.32) |
| **F** | 0.52 (0.39 − 0.45) | 0.51 (−0.24 − 0.81) | 0.56 (0.45 − 0.63) | 0.45 (0.01 − 0.63) | 0.23 (0.08 − 0.27) | 0.63 (0.53 − 0.73) | 0.09 (−0.10 − 0.13) | 0.37 (0.15 − 0.38) |

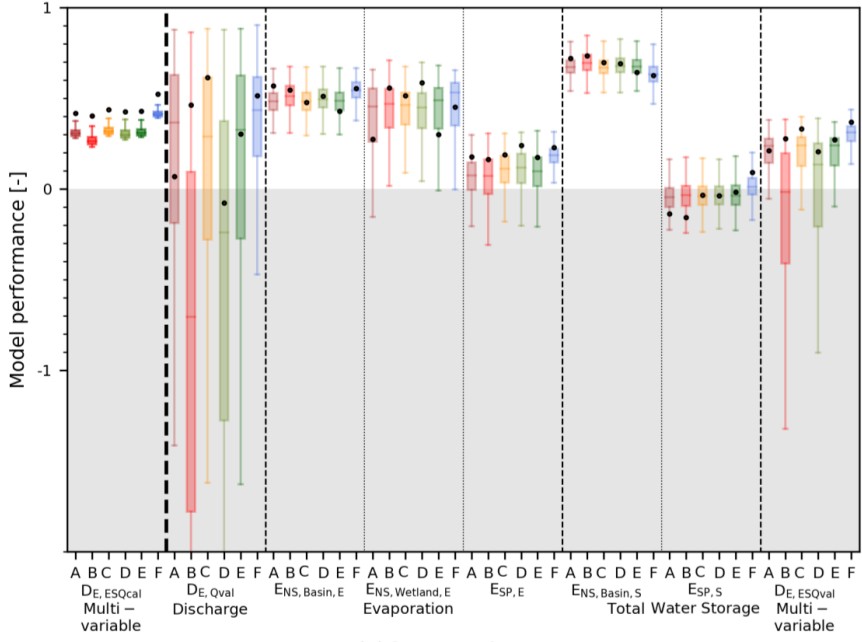

**Figure 8: Model performance with respect to discharge, evaporation and storage for all models. The model is calibrated to all fluxes simultaneously ($D_{E,ESQcal}$, darker boxplots in the first column) and evaluated with respect to each flux individually (lighter boxplots). The dots represent the model performance using the "optimal" parameter set and the boxplot the range of the best 5% solutions both according to $D_{E,ESQcal}$. The following performance metrics were used: 1) discharge using the overall model performance metric ($D_{E,Qval}$), 2) evaporation temporally basin-average ($E_{NS,Basin,E}$), 3) evaporation temporally wetland areas only ($E_{NS,Wetland,E}$), 4) evaporation spatially ($E_{SP,E}$), 5) storage temporally basin-average ($E_{NS,Basin,S}$), 6) storage temporally wetland areas only ($E_{NS,Wetland,S}$), 7) storage spatially ($E_{SP,S}$), and 8) the combination of evaporation, storage and discharge (combined metric $D_{E,ESQval}$).**

**Figure 9: Range of model solutions for Models A to F. The left panel shows the hydrograph and the right panel the flow duration curve of the recorded (black) and modelled discharge: the line indicates the solution with the highest calibration objective function with respect to multiple variables ($D_{E,ESQcal}$) and the shaded area the envelope of the solutions retained as feasible. The data in the grey shaded area were used for validation.**



**Figure 10: Range of model solutions for Models A to F. The left panel shows the time series and the right panel the duration curve of the recorded (black) and modelled normalised evaporation for wetland dominated areas: the line indicates the solution with the highest calibration objective function with respect to multiple variables ($D_{E,ESQcal}$) and the shaded area the envelope of the solutions retained as feasible. The data in the grey shaded area were used for validation.**





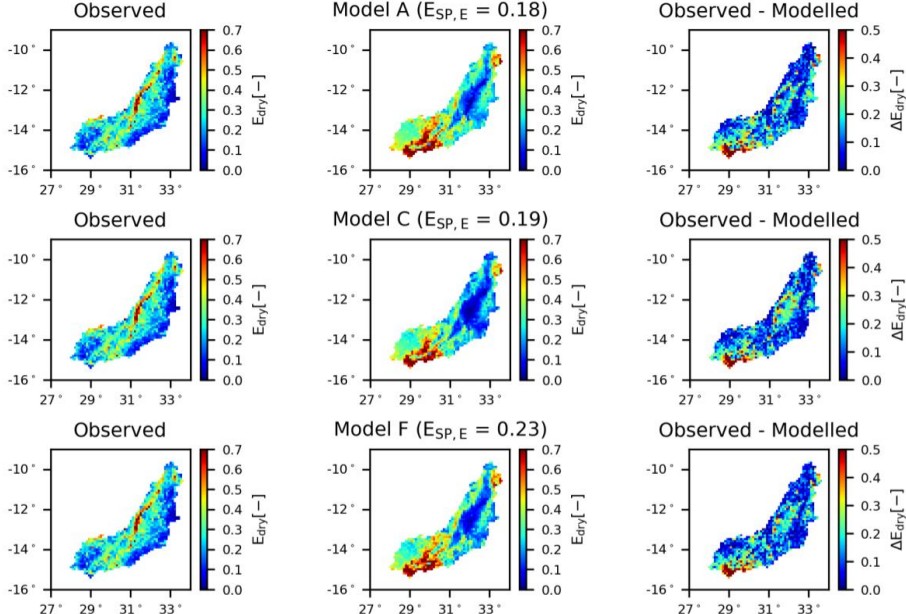

**Figure 11:** Spatial variability of the normalised total evaporation for Models A, C and F averaged over all days within the dry season. The left panel shows the observation according to WaPOR data, the middle panel the model result using the "optimal" parameter set with respect to multiple variables ($D_{E,ESQcal}$), and the right panel the difference between the observation and model.

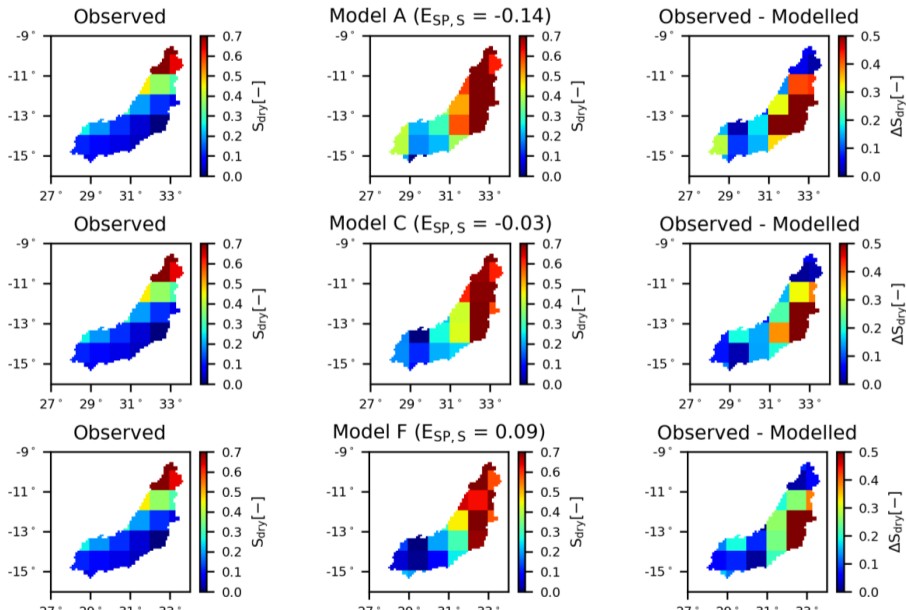

**Figure 12:** Spatial variability of the normalised total water storage anomalies for Models A, C and F averaged over all days within the dry season. The left panel shows the observation according to GRACE data, the middle panel the model result using the "optimal" parameter set with respect to multiple variables ($D_{E,ESQcal}$), and the right panel the difference between the observation and model.





### 4.1.3 Model deficiencies

Regardless of the calibration strategy, the benchmark model failed in particular to adequately reproduce evaporation dynamics in wetland dominated areas. During the dry seasons, the modelled evaporation decreased rapidly to zero in contrast to the observations (Figures 5 and 10). Partly as a consequence of that, the spatial pattern of evaporation was captured poorly

as illustrated in Figures 6 and 11. Apart from the wetlands, the modelled average dry season evaporation was also extremely low in the centre of the basin which did not correspond with the satellite observations. At the same time, the evaporation was significantly overestimated in the southern part of the basin. Also the spatial pattern in total water storage anomalies were poorly represented since the model significantly overestimated storage anomalies in large parts of the basin (Figures 7 and 12).

### 4.2 Adding groundwater upwelling (Models B, C and D)

In the benchmark model (Model A), there was no groundwater upwelling into the wetlands and floodplains around the river channels, similar to many distributed conceptual hydrological models (eg. Samaniego et al., 2010; Bieger et al., 2017). However, according to field and satellite-based observations, wetland areas remain moist at the end of the dry season while the remaining areas of the basin become very dry. Given the low elevation of these wetlands above rivers, it is plausible to

assume that groundwater from higher parts of the catchment is pushed up into the unsaturated root zone of these wetlands. As a result, water deficits in the unsaturated zone are partly replenished by upwelling groundwater. It thereby can sustain relatively elevated levels of moisture, available for plant transpiration long into the dry season.

To improve the representation of evaporation in the model, the process of upwelling groundwater ($R_{GW}$) was added to the model. In principle, it was assumed that the upwelling groundwater is regulated by the head difference between upland

groundwater and the groundwater in the wetland. As this information was not available, due to the lack of continuous gradients in the type of model used (Hrachowitz and Clark, 2017), this was done in a simplified way. In three alternative formulations of this hypothesis, the upwelling groundwater was made (1) a linear function of the water content in the unsaturated reservoir (Model B, Eq.9), (2) a linear function of the water content in the slow responding reservoir (Model C, Eq.10) and (3) a non-linear function of the water content in the slow responding reservoir (Model D, Eq. 11). In other words,

in Model B the groundwater upwelling was driven by the water deficit in the unsaturated zone, hence the lower the water content in the unsaturated zone, the higher the groundwater upwelling. In Models C and D, the groundwater upwelling was driven by the water content in the slow responding reservoir, the groundwater system, such that the higher the water content in the slow responding reservoir, the higher the groundwater upwelling. As a result of the non-linear relation between the groundwater upwelling and the water content in the slow responding reservoir in Model D, the groundwater upwelling

increased the most under dry conditions and less under wet conditions. In Models B – D, the groundwater upwelling flowed into the unsaturated zone until it was saturated, hence until its maximum $S_{u,max}$ was reached (Eq.12). Model B required one additional calibration parameter, Model C two and Model D three (Tables 2 and 3).

### 4.2.1 Discharge based calibration

Following the first calibration strategy, the performances of Models B – D with respect to discharge did not improve

significantly for the calibration period ($D_{E,Qcal} = 0.75 – 0.79$) compared to Model A, regardless of the model (Table 6, Figures 3 and S2). For the validation period, Models B and D experienced a pronounced reduction of their ability to adequately reproduce the discharge signatures with $D_{E,Qval} = 0.08$ and $-1.7$, respectively, since the flows were mostly underestimated (Figure S2). On the other hand, Model C showed significant improvements with $D_{E,Qval} = 0.81$. With respect to the evaporation from wetland dominated areas, the largest improvements were found for Model D ($E_{NS,Wetland,E} = 0.41$)

where the evaporation did not drop rapidly to zero anymore even though it was still significantly underestimated in the dry season (Figure S4). But this came at the cost of decreased simulations of all remaining variables (Table 6, Figure 3), hence





the discharge, basin-average evaporation and total water storage and their spatial pattern (Figures S2 – S7). For example Figure S6 illustrates the poorly simulated temporally-averaged dry season evaporation for Model D which was higher in wetland areas (centre of the basin) compared to the surrounding areas which was not observed in the satellite based

observations. For Models B and C, the model performances with respect to the remaining variables remained comparable to Model A or even decreased as can be seen in Table 6 and Figure 3. As a result, when considering all variables simultaneously, Model C performed the best with $D_{E,ESQval} = 0.32$.

### 4.2.2 Multi-variable calibration

Following the second calibration strategy, Model C experienced the largest increases compared to Model A in its ability to

describe features of discharge with $D_{E,Qval} = 0.61$, while Model D decreased the most to $D_{E,Qval} = -0.08$ with the high flows being overestimated and low flows underestimated (Table 7, Figures 8 and 9). With this calibration strategy, large improvements were observed in the reproduction of the evaporation from wetland dominated areas for all three Models B – D, especially for Model D with $E_{NS,Wetland,E} = 0.59$ where the evaporation was simulated well even during the dry season as it did not decrease rapidly to zero in the dry season compared to Model A (Figure 10). For Models C and D, the spatial pattern

in evaporation and total water storage anomalies improved, albeit moderately (Table 7) as large areas were still under- or overestimated (Figures S10 and S11), whereas it decreased slightly for Model B. For all Models B – D, the basin-average temporal dynamics in evaporation and total water storage anomalies remained similar or decreased slightly (Table 7, Figures S8 and S9). Overall, when considering the model performance with respect to all variables simultaneously, Model C showed the highest performances with $D_{E,ESQval} = 0.33$.

### 4.2.3 Model deficiencies

According to the results, the representation of evaporation strongly benefitted from including upwelling groundwater as function of the water content in the slow responding reservoir (Eq.10, Model C) especially for the second calibration strategy. The incorporation of this flux resulted in increased levels of water supply to the unsaturated zone of wetlands to sustain higher levels of transpiration throughout the dry periods (Figure 10). But even though the evaporation increased

during dry periods, it was still underestimated especially towards the end of the dry season due to too large groundwater upwelling depleting the slow responding reservoir. The major weakness of the model remained its very limited ability to represent the spatial pattern in evaporation as there were several local clusters of considerable mismatches, both over- and underestimating observed evaporation. This was clearly visible for example in the centre and southern part of the basin (Figure 11). Also the spatial pattern in the total water storage anomalies remained poorly represented, in spite of some

improvements compared to Model A, as they were considerably overestimated in the northern parts of the basin (Figure 12). This could be a result of deficiencies in the hydrological models or in the satellite-based observations.

### 4.3 Discretizing the groundwater system (Models E and F)

In all above models, the groundwater layer was simulated as a single lumped reservoir assuming equal groundwater availability throughout the entire basin. As groundwater processes can occur on relatively large spatial scales, this

assumption may be valid for small- or mesoscale catchments, but not necessarily for larger basins such as the Luangwa basin. This may partly be responsible for the deficiency of all above models to meaningfully reproduce the spatial pattern of the total water storage. Taking Model C as a basis for further model adaptations, two more alternative model hypothesis were formulated. In both models the slow responding reservoir, representing the groundwater, was spatially discretized. For Model E, the reservoir was split into four units with an area of 15,396 – 47,239 km$^2$ each containing four to six different

GRACE cells (see Figure 1A). In contrast, Model F was formulated with a completely distributed slow reservoir at the resolution of the remaining parts of the model, i.e. 10 x 10 km$^2$. In Models E and F, the slow reservoir timescales $K_s$



remained constant throughout the basin to limit the number of calibration parameters. Models E and F did not require additional calibration parameters. See Tables 2 and 3 for the corresponding model equations and calibration parameter ranges.

### 4.3.1 Discharge based calibration

Following the first calibration strategy, the calibrated and validated model performance with respect to discharge did not change significantly for Model E compared to Model C. For Model F on the other hand, the calibrated model performance increased to $D_{E,Qcal} = 0.91$ (Table 6, Figures 3 and S2), but during validation it decreased to $D_{E,Qval} = 0.52$ compared to Model C as a result of overestimated high flows (Figure S2). In other words, the discharge simulation was only affected

when applying a fully distributed groundwater system (Model F). Also the simulated dynamics of the evaporation improved for Model F, especially for wetland dominated areas ($E_{NS,Wetland,E} = 0.56$, Table 6) even though it remained significantly underestimated during the dry season (Figure S4). But for both models, no improvements in the spatial pattern of evaporation can be observed with $E_{SP,E} = 0.05$ and -0.03 for Models E and F, respectively. As shown in Figure S6, for Model E and F the temporally-averaged dry season evaporation was very low in the centre of the basin compared to the remaining part of the basin in contrast to the satellite-based observations. The spatial pattern of total water storage anomalies were at

least slightly better mimicked by Model F with $E_{SP,S} = 0.08$ (Figure S7), which, in turn, came at the price of a poorer reproduction of the temporal dynamics of the basin-averaged total water storage anomalies ($E_{NS,Basin,S} = 0.66$, Figure S5).

### 4.3.2 Multi-variable calibration

Including multiple variables in the calibration process did not improve the representation of the hydrological response with

respect to discharge for Models E and F compared to Model C with $D_{E,Qval} = 0.30$ and 0.51, respectively (Table 7, Figures 8 and 9). For both models, the flows were underestimated during low flows and overestimated during high flows (Figure 9). Also the evaporation from wetland dominated areas did not improve for both models as it decreased rapidly in the dry season (Figure 10). On the other hand, the spatial pattern in the evaporation was slightly better mimicked for Model F ($E_{SP,E} = 0.23$), but still at low performance levels similar to Models A – D with large areas still being under- or overestimated (Figure S10).

Slight improvements could be observed though for the representation of spatial pattern in total water storage in Models F ($E_{SP,S} = 0.09$, Figure S11), albeit modestly. Overall, when considering the model performance with respect to all variables simultaneously, Model F showed the highest performances with $D_{E,ESQval} = 0.37$.

### 4.3.3 Model deficiencies

Applying the second calibration strategy, Model F poorly reproduced the evaporation from wetlands (Figure 10) since the water availability for evaporation decreased rapidly in the dry season due to the limited water availability in the slow responding reservoir. This was a direct result of the limited connectivity in the distributed groundwater system within the basin and very likely points to the presence of contiguous groundwater systems extending beyond the modelling resolution that sustain dry season evaporation in wetlands. Strikingly, discretizing the groundwater basin only had limited effects on the

spatial pattern in evaporation and total water storage anomalies. Despite their limited improvements, they remained poorly captured as several local clusters were over- and underestimated (Figures 11 and 12).





## 5. Discussion

As illustrated in the previous sections, satellite-based evaporation and storage anomaly data were used in an attempt to (1) iteratively improve a benchmark model structure and 2) identify parameter sets with which the model can simultaneously

reproduce the temporal dynamics as well as the spatial pattern of multiple flux and storage variables.

The results suggested that among the tested models, Models C and F provided the overall best representation of the hydrological processes in the Luangwa basin, following the first and second calibration strategy respectively. The addition of upwelling groundwater alone (Model C) significantly improved the discharge simulations during validation regardless of the calibration strategy and the simulation of evaporation from wetland areas following the second calibration strategy.

Discretizing the slow responding reservoir (Model F) reached reasonable overall performance levels, i.e. $D_{E,ESQval}$, when calibrating on discharge and its signatures only (Figure 3), with improved simulations of evaporation from wetland areas. But calibrating on multiple variables proved instrumental as it allowed to significantly improve the spatial pattern of the evaporation, while maintaining high levels for the other performance criteria (Figure 8). In general it could also be observed that a further discretization of the model lead to a better representation of the system especially with respect to the spatial

pattern. Nevertheless, none of the tested models could adequately reproduce the observed spatial pattern in evaporation and total water storage anomalies which could be a result of model deficiencies or uncertainties in the satellite-based observations of the spatial pattern.

A potential reason for the models' problems to meaningfully describe the spatial pattern of the evaporation was in this study the use of the same parameters within a specific HRU in different model grid cells as also observed in previous studies

(Stisen et al., 2018). As a result, the simulated spatial pattern was strongly influenced by the catchment classification method into distinct HRUs. In this study, the catchment was classified merely on the basis of topography into flat, sloped and wetland areas, whereas ecosystem diversity could also be considered as an additional layer in the classification. The poor representation of the spatial pattern in total water storage was also partly linked to that. Another likely reason is the absence of lateral exchange of sub-surface water between model grid cells in the tested models, as contiguous groundwater bodies of

varying but unknown spatial scale will shape water transfer through the landscape in the real world which remain unaccounted for in the model.

In addition, each of the applied data sources have their own uncertainties and bias. These include uncertainties in observed discharge due to rating curve uncertainties (Westerberg et al., 2011; Tomkins, 2014; Domeneghetti et al., 2012) and limited data availability, in precipitation data, often as a result of poorly capturing mountainous regions or extreme events on small

scales (Kimani et al., 2017; Dinku et al., 2018; Le Coz and van de Giesen, 2019; Hrachowitz and Weiler, 2011), in estimates of total water storage anomalies as a result of data (post-) processing including data smoothing using a radius of for example 300 km affecting the spatial variability on basin scale (Landerer and Swenson, 2012; Blazquez et al., 2018) and in evaporation data due to model, input data and parameter estimation uncertainties (Zhang et al., 2016). In general satellite products are a result of models that are prone to uncertainties related to the input data or model conceptualisation. In the

ideal situation, the data would be validated with field measurements to assess the error magnitude. However, this was not possible due to data limitations.

The results in this study were sensitive to the choice of performance metrics with respect to the individual variables (discharge, evaporation and total water storage) and all variables combined. For instance the overall model performance measure $D_{E,ESQval}$ (Eq.32) was strongly influenced by the validated discharge model performance $D_{E,Qval}$ due to its large range

and variation between models compared to the remaining variables where the range was smaller and similar for all models (Figure 8). As a result, the overall model performance measure might not reflect each variable equally well which affected the choice of best performing model.

Reflecting the results of previous studies, this study found that calibrating to multiple variables including the spatial pattern improved the simulation of the evaporation and storage with some trade-off in the discharge simulation depending on the


model structure (Herman et al., 2018; Rientjes et al., 2013; Stisen et al., 2011; 2018; Demirel et al., 2018; Dembélé et al., 2020). But in contrast and additional to previous studies, this study also provided an example, illustrating that spatial data, here evaporation and total water storage, can contain relevant information to diagnose model deficiencies and to therefore enable step-wise model structural improvement. Previous studies have largely relied on discharge observations to improve model structures (Fenicia et al., 2016; Hrachowitz et al., 2014) and only few studies used satellite data (Roy et al., 2017)

even though it provides valuable information on the internal processes temporally and spatially which is not available with discharge data alone (Rakovec et al., 2016; Daggupati et al., 2015). Roy et al. (2017) observed that the simulated evaporation according to the spatially lumped model HYMOD (HYdrological MODel) rapidly dropped to zero in contrast to the satellite product GLEAM (Global Land Evaporation Amsterdam Model) in the Nyangores river basin in Kenya. They improved this simulated evaporation while maintaining good discharge performances by modifying the corresponding equation in

HYMOD such that it was a function of the soil moisture. While here we focussed on upwelling groundwater and spatial discretization, a promising avenue for future studies may be to evaluate the incorporation of simple formulations of subsurface exchange fluxes between model grid cells. Similarly, a further discretization of HRUs into different land cover and ecosystem types may be worthwhile.

## 6. Conclusion

The objective of this paper was to explore the added value of satellite-based evaporation and total water storage anomaly data to increase the understanding of hydrological processes through step-wise model structure improvement and model calibration for large river systems in a semi-arid, data scarce region. For this purpose, a distributed process-based hydrological model with sub-grid process heterogeneity for the Luangwa River basin was developed and iteratively adjusted. The results suggested that (1) the benchmark model (Model A) calibrated with respect to discharge simulated the discharge

well, and also the basin-average evaporation and total water storage anomalies, but poorly captured the evaporation for wetland dominated areas and the spatial pattern of evaporation and total water storage anomalies. (2) Testing five further alternative model structures (Models B – F), it was found that among the tested model hypotheses Model F, allowing for upwelling groundwater from a distributed representation of the groundwater reservoir and (3) simultaneously calibrating the model with respect to multiple variables, i.e. discharge, evaporation and total water storage anomalies, provided the best

representation of all these variables with respect to their temporal dynamics and spatial pattern, except for the basin-average temporal dynamics in the total water storage anomalies. It was shown that satellite-based evaporation and total water storage anomaly data are not only valuable for multi-criteria calibration, but can play an important role in improving our understanding of hydrological processes through diagnosing model deficiencies and step-wise model structural improvement.

**Abbreviations**

|       |                                                              |
|-------|--------------------------------------------------------------|
| CHIRPS | Climate Hazards Group InfraRed Precipitation with Station   |
| CMRSET | CSIRO MODIS Reflectance Scaling EvapoTranspiration          |
| CRU    | Climatic Research Unit                                       |
| CSIRO  | Commonwealth Scientific and Industrial Research Organisation |
| FAO    | Food and Agriculture Organization                            |
| GEOS   | Goddard Earth Observing System Model                         |
| GMTED  | Global Multi-resolution Terrain Elevation Data               |
| GRACE  | Gravity Recovery and Climate Experiment                      |
| HRU    | Hydrological Response Unit                                   |



| 595 | MERRA | Modern-Era Retrospective analysis for Research and Applications |
| | MODIS | Moderate Resolution Imaging Spectroradiometer |
| | NDVI | Normalized Difference Vegetation Index |
| | SSEBop | operational Simplified Surface Energy Balance |
| | WaPOR | Water Productivity Open Access Portal |

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
