# Peer review of "Learning from satellite observations: increased understanding of catchment processes through stepwise model improvement"

_Hydrology and Earth System Sciences, 2020_

## Referee Comment (RC1) · Anonymous Referee #1 · 14 Jun 2020

This manuscript reports on a comprehensive calibration and validation experiment of a hydrological model at large spatial scales. The value of this manuscript is less on learning on a particular model, on the hydrology of a river basin, or on how a suitable and well performing hydrological model should look like for this particular region, but its value is much more on presenting a well-defined procedure of a step-wise multi-variable and multi-criterial calibration scheme towards improving model structure. The study illustrates the use of even coarse and uncertain remote sensing data, including the spatial patterns of state and flux variables (here water storage variations and evapotranspiration) in the calibration and model adjustment approach. In this respect, it provides a valuable example and guideline for other studies in future. I therefore

recommend its publication in HESS after considering some comments as listed below.

1) A thematic / scientific drawback of the study is that a significant improvement of the spatial patterns of simulated ET and storage anomalies could not be achieved within the set of model modifications tested here, in spite of some increase in the performance criterion. In particular, the pattern of areas with high ET in the remote sensing product could not be reproduced by the model. The authors argue that missing lateral sub-surface flow between modelling units could be a reason for this. Can a related modification of the model structure additionally be tested? A more convincing outcome in this direction could also be of benefit for the paper as a whole in demonstrating the value of the multi-criterial calibration approach on spatial patterns.

2) In this respect, the authors discuss the dominance of the discharge performance criterion within the overall performance measure that was used for calibrating against all variables and criteria. Has the ability of the model to reproduce that spatial ET patterns been tested with varying weights among the different criteria in the overall measure, or for single-criterion calibration the ET patterns only? The (in)ability of the model to represent this feature and the trade-offs relative to other criteria could be another good indicator of structural model deficits.

3) The satellite-based data product used here for calibration and validation is an actual evapotranspiration product, isn't it? I suggest to change the term evaporation to ET throughout the manuscript.

4) line 114: "In this study, the long-term bias between the discharge, evaporation (WaPOR) and total water storage anomalies (GRACE) was corrected by multiplying the evaporation with a correction factor of 1.08 to close the long-term water balance." What about precipitation? Its amount is required to close the water balance.

5) Figure 5, caption: "Range of model solutions for Models A to F." This should read "Model A" only.

6) Figure 7 and 12: "Spatial variability of the normalised total water storage anomalies for Model A averaged over all days within the dry season."

7) line 403: "... since the model significantly overestimated storage anomalies in large parts of the basin." This statement can be misleading. After normalisation with Eq. 33, a higher value of the model compared to GRACE indicates that the negative storage anomaly of the model is less pronounced than the one of GRACE because the averaging period considered here is the dry season?

8) For model calibration, a simple Monte-Carlo parameter sampling strategy is applied in spite of the fact that there are effective multi-criterial calibration methods around that can be expected to result in parameters sets with higher model performance than obtained here, such as Borg or other evolutionary algorithms. While I am not necessarily recommending to use such algorithms for the present study as its aim is rather on comparative model evaluation and development than on pure parameter optimization, the authors may explain their choice.

---

## Author Comment (AC1) · 7 Jul 2020

We thank the reviewer for his/her interest in our work and for the thoughtful and detailed feedback provided.

*Comment:*

*This manuscript reports on a comprehensive calibration and validation experiment of a hydrological model at large spatial scales. The value of this manuscript is less on learning on a particular model, on the hydrology of a river basin, or on how a suitable and well performing hydrological model should look like for this particular region, but its value is much more on presenting a well-defined procedure of a step-wise multi-variable and multi-criterial calibration scheme towards improving model structure. The study illustrates the use of even coarse and uncertain remote sensing data, including the spatial patterns of state and flux variables (here water storage variations and evapotranspiration) in the calibration and model adjustment approach. In this respect, it provides a valuable example and guideline for other studies in future. I therefore recommend its publication in HESS after considering some comments as listed below*

Reply:

We highly appreciate this positive assessment of our manuscript. We will in the following address all specific comments in detail.

*Comment:*

*A thematic/scientific drawback of the study is that a significant improvement of the spatial patterns of simulated ET and storage anomalies could not be achieved within the set of model modifications tested here, in spite of some increase in the performance criterion. In particular, the pattern of areas with high ET in the remote sensing product could not be reproduced by the model. The authors argue that missing lateral sub-surface flow between modelling units could be a reason for this. Can a related modification of the model structure additionally be tested? A more convincing outcome in this direction could also be of benefit for the paper as a whole in demonstrating the value of the multi-criterial calibration approach on spatial patterns.*

Reply:

This is indeed a very important point raised by the reviewer. It is true that even after rather extensive model testing and adaption, the representation of spatial pattern did only slightly improve. We also think that it is not implausible to assume that lateral exchange between the model units may explain some of these model deficiencies. Adding such lateral exchange to the model is of course possible, in principle. However, it is not a trivial thing to do in a meaningful way. We believe that this is in itself is a major research topic which warrants several in-depth research papers on its own and which cannot be done as a mere additional hypothesis in this analysis. The underlying reason for this is partly implicit in the nature of the model type used here and partly in the data that are available with current observation technology. Lateral exchange fluxes are (as any fluxes) driven by the interplay between continuous gradients and resistances. Conceptual type models are based on simplified expressions that mimic gradients *within* a model domain only. If such a model is implemented in a spatially distributed way, the individual model domains are the model grid cells. Gradients are thus only defined *within* these grid cells, but not across them. Thus, the head difference between adjacent model grid cells is undefined. In the absence of such gradients it therefore remains unknown between which grid cells such lateral

exchange fluxes occur, into which direction and at which rates. As a consequence, these fluxes can only be expressed on basis of free calibration parameters. Depending on the degree of spatial discretization at the very least 4 additional free calibration parameters (Model E) would here be needed to represent exchange fluxes between the model grid cells. This does not yet include potential exchange fluxes of each grid cell with adjacent grid cells outside the Luangwa basin. In comparison, the fully distributed Model F would even require at least 30 additional calibration parameters. In the model calibration process, these additional parameters and the associated increase in the degree-of-freedom of the model, will very likely lead to improved model performances. This may even extend to the model validation period. Yet, the inclusion of such processes will not be warranted by the available data, as we will have no means of testing whether the additional calibration parameters and the associated exchange fluxes are physically plausible. We may end up with a model that features nice performance metrics for calibration and potentially also for validation, but in which water may flow against real-world elevation and/or pressure gradients or, to express it in a pointed way, water may flow uphill. These unspecified boundary fluxes across grid cells are at the core of the closure problem (Beven, 2006) and touch on the limits of what can be done in hydrology with our current observational technology and the available data. We will discuss this in more detail in the revised manuscript.

*Comment:*

*In this respect, the authors discuss the dominance of the discharge performance criterion within the overall performance measure that was used for calibrating against all variables and criteria. Has the ability of the model to reproduce that spatial ET patterns been tested with varying weights among the different criteria in the overall measure, or for single-criterion calibration the ET patterns only? The (in)ability of the model to represent this feature and the trade-offs relative to other criteria could be another good indicator of structural model deficits.*

Reply:

We agree that, as discussed in the paper and mentioned by the Reviewer, the discharge performance criterion had a dominant influence on the multi-criteria model performance. This was especially visible when comparing different models with each other (see Figures 3 and 8 in the manuscript) and could indeed be one of the reasons for the poor simulation of the spatial pattern in the evaporation. To test this, the models were also calibrated with respect to the spatial pattern in the evaporation only. This did improve this variable, but only to a certain extent (Figure 1 below). We discussed this in Section 5 of the original manuscript, where we emphasized that it is plausible to assume that the poor simulation of the spatial pattern was more likely a result of using the same parameters within a specific HRU for all grid cells throughout the basin as also observed in previous studies (Stisen et al., 2018). We will expand the discussion of that issue in the revised manuscript and provide Figure 1 below as supplementary material to the manuscript.

[Figure]

**Figure 1: Spatial variability of the normalised total evaporation for Model F averaged over all days within the dry season. The left panel shows the observation according to WaPOR data, the middle panel the model result using the "optimal" parameter set with respect to the spatial pattern in the evaporation ($E_{SP,E}$), and the right panel the difference between the observation and model.**

*Comment:*

*The satellite-based data product used here for calibration and validation is an actual evapotranspiration product, isn't it? I suggest to change the term evaporation to ET throughout the manuscript.*

Reply:

Thank you for pointing this out. This study used the satellite product WaPOR for calibration and validation. This product describes the actual total evaporation as the sum of the individual components interception, evaporation, soil evaporation and plant transpiration (FAO, 2018). While many studies use the term "evapotranspiration" to describe the combination of different evaporation processes, other studies use the term "evaporation" as overarching term. The FAO defines "evapotranspiration" as the sum of evaporation from different surfaces and transpiration from plants. However, as argued by Savenije (2004), interception and soil evaporation, on the one hand, are functionally completely different processes than transpiration, on the other hand. While the latter is constrained by moisture in the root zone and contains a biological component of water released by stomata before the physical processes of evaporation from the surfaces of leaves occurs, interception and soil evaporation are purely physical processes. We therefore prefer to keep the term "evaporation" as an overarching term as, strictly spoken, there is no single process that can be referred to as "evapotranspiration" (Savenije, 2004; Brutsaert, 1982; 2005).

*Comment:*

*line 114: "In this study, the long-term bias between the discharge, evaporation (WaPOR) and total water storage anomalies (GRACE) was corrected by multiplying the evaporation with a correction factor of 1.08 to close the long-term water balance." What about precipitation? Its amount is required to close the water balance.*

Reply:

In general, an open long-term water balance could indeed be a result of uncertainties in precipitation, evaporation and/or discharge. As a result of limited ground observations, it was not possible to validate the satellite-based observations to correct for errors such as bias. In this study, the hydrological model and satellite-based evaporation product WaPOR used the same precipitation product CHIRPS (FAO, 2018). As a result, any bias between modelled and satellite-based evaporation cannot be a result of the precipitation (even though it

could be a reason for the water balance non-closure), but can be a result of different underlying methodologies. That is why we chose to only correct the evaporation.

As simple comparison, the model was run with a random parameter set adjusting 1) the observed evaporation (factor 1.08) and 2) the precipitation only (factor 0.93). The modelled evaporation decreased slightly in Scenario 2 compared to Scenario 1 as it decreased with an average of 0.1 mm/d and a maximum of 0.5 mm/d (Figure 2 here below). The model performance with respect to the temporal variation in the evaporation was also very similar to each other with $E_{\text{NS,Basin,E}} = 0.65$ for Scenario 1 and $E_{\text{NS,Basin,E}} = 0.66$ for Scenario 2 since normalised values were used as explained in the paper.

[Figure]

**Figure 2: Modelled evaporation for one random simulation when adjusting the 1) observed evaporation (corresponding to $E_1$), or 2) precipitation only (corresponding to $E_2$) to close the long term water balance. The red line in the right panel indicates the 1:1 line.**

*Comment:*

*Figure 5, caption: "Range of model solutions for Models A to F." This should read "Model A" only.*

Reply:

Thank you for pointing this out. This will be corrected in the revised version of the manuscript.

*Comment:*

*Figure 7 and 12: "Spatial variability of the normalised total water storage anomalies for Model A averaged over all days within the dry season."*

Reply:

In Figures 7 and 12, the spatial variability of the normalised total water storage anomalies was visualised for Model A. In Figure 12, also Models C and F were included. The difference between both figures is that in Figure 7, the "optimal" parameter set was selected based on discharge data only ($D_{\text{E,Qcal}}$), while in Figure 12 multiple-variables were used to for this purpose ($D_{\text{E,ESQcal}}$). This was indicated in the second part of the figure caption for both figures. We will further clarify this in the revised manuscript.

*Comment:*

*line 403: "...since the model significantly overestimated storage anomalies in large parts of the basin." This statement can be misleading. After normalisation with Eq.33, a higher value of the model compared to GRACE indicates that the negative storage anomaly of the model is less pronounced than the one of GRACE because the averaging period considered here is the dry season?*

Reply:

Thank you for pointing this out. This statement can indeed be confusing. With respect to the spatial pattern, spatially normalised values were compared with each other instead of absolute values. As a result, a higher normalised model value compared to the observation does not necessarily mean the actual (non-normalised) model value was also higher. However, it does mean the simulation results in this cell/region were high relative to the remaining of the basin compared to the observation.

To illustrate this, the simulated and observed dry season total water storage anomalies was visualised considering their normalised values (Figure 3 here below) and actual values (hence non-normalised, Figure 4 here below). Figure 3 shows that for Model A several cells have higher normalised values compared to the observation (e.g. the marked cell), while the actual modelled values are lower than the observation as shown in Figure 4 (please note the scale bar in Figure 4 is different for the observation and model). However, both Figures 3 and 4 show similar spatial pattern. For example in Figure 4, the marked cell in the modelled map shows a high value compared to the remaining of the basin, which was also the case in Figure 3. As a result, the spatial pattern was preserved when normalising the maps, also when calculating only with negative values as is the case when considering the total water storage map averaged over the dry season.

We will reformulate this statement to highlight this and avoid any confusion.

[Figure]

**Figure 3: Spatial variability of the *normalised* total water storage anomalies for Model A averaged over all days within the dry season. The left panel shows the observation according to GRACE data, and the right panel the model result using the "optimal" parameter set with respect to discharge ($D_{E,Qcal}$).**

[Figure]

**Figure 4: Spatial variability of the total water storage anomalies (*not normalised*) for Model A averaged over all days within the dry season. The left panel shows the observation according to GRACE data, and the right panel the model result using the "optimal" parameter set with respect to discharge ($D_{E,Qcal}$).**

*Comment:*

*For model calibration, a simple Monte-Carlo parameter sampling strategy is applied in spite of the fact that there are effective multi-criterial calibration methods around that can be expected to result in parameters sets with higher model performance than obtained here, such as Borg or other evolutionary algorithms. While I am not necessarily recommending to use such algorithms for the present study as its aim is rather on comparative model evaluation and development than on pure parameter optimization, the authors may explain their choice.*

Reply:

Thank you for this comment. There are indeed many multi-criteria calibration methods that can be very useful to find the "optimal" parameter set and associated posterior parameter distributions. However, the goal of this study was to explore the information content of multiple variables using multiple model evaluation criteria for step-wise model structure development and calibration. For this purpose, it was important to use the same parameter sets for all models as common starting point to rule out the effect of different parameter sets. This was efficiently possible with the Monte-Carlo parameter sampling strategy, which, in addition also allowed a relatively straight-forward and intuitive interpretation and communication of the results. We will add an explanation of this choice in the revised manuscript.

**References**

Beven, K.: Searching for the Holy Grail of scientific hydrology: Qt=(S, R, Δt)A as closure, Hydrol. Earth Syst. Sci., 10, 609-618, 10.5194/hess-10-609-2006, 2006.
Brutsaert, W.: Evaporation into the atmosphere: Theory, history, and applications, Springer, Dordrecht, Heidelberg, London, New York, 299 pp., 1982.
Brutsaert, W.: Hydrology: An Introduction, Cambridge University Press, Cambridge, 2005.
FAO: WaPOR Database Methodology: Level 1. Remote Sensing for Water Productivity Technical Report: Methodology Series, in, FAO, Rome, 72, 2018.
Savenije, H. H. G.: The importance of interception and why we should delete the term evapotranspiration from our vocabulary, Hydrological Processes, 18, 1507-1511, 10.1002/hyp.5563, 2004.
Stisen, S., Koch, J., Sonnenborg, T. O., Refsgaard, J. C., Bircher, S., Ringgaard, R., and Jensen, K. H.: Moving beyond run-off calibration—Multivariable optimization of a surface–subsurface–atmosphere model, Hydrological Processes, 32, 2654-2668, 10.1002/hyp.13177, 2018.

---

## Referee Comment (RC2) · Anonymous Referee #2 · 25 Aug 2020

General comments:

The manuscript is very well written and nicely illustrated. It deals with an interesting topic of model structure and gaining information from satellite data in a data sparse basin.

It makes the paper less interesting that there are essentially no major changes to the simulated spatial patterns of ET across Model A-F (e.g. Figure 11) and that the general simulated spatial pattern does not resemble the observed in any way. It seems that you are not addressing the most important issues in your set of alternative models (B-F). The general interest of the manuscript would increase greatly if some of your

hypothesis would at least produce a different pattern from Model A.

The simulated spatial pattern will be a reflection of both model structure and parametrizations scheme, in my experience mainly the latter. Therefore, I strongly suggest that you add a set of model setups that reflect different spatial parameterization schemes. In your discussion you address this limitation nicely, but I also feel that the manuscript would benefit greatly from an additional analysis illustrating the importance of model parameterization and parameter distribution on the simulated spatial patterns. Basically, even the most sophisticated model structure cannot be expected to reproduce a correct spatial pattern without a sound, flexible and spatially explicit parametrizations scheme.

I think you can logically add such an analysis to your manuscript in line with the idea of learning from satellite observations, by letting the observed spatial patterns guide your spatial parametrization approach. Such a parametrization scheme could also include transfer functions or simple spatial relations to known variables such as elevation, slope, soiltype, LAI etc.

In sections 3.1.2 and 3.1.3 it is unclear why the different structural changes were applied. The title suggests that you are learning from satellite data, but it is not clear to me how you learn and how you used the satellite data to make new hypothesis about model structure. It is mentioned several times that you diagnose model deficiencies, however it is unclear to me how this is done. I believe this should be elaborated in a revised manuscript.

**Specific comments:**

An issue with the use of the SPAEF metric for the water storage anomaly might be, that the histogram component of the metric, might not be so meaningful when applied to the coarse spatial resolution of 1 deg., with very few grids. You could look into this by examining the three components of the metric separately. I do not suggest to put this analysis in the paper, but it might be mentioned in a discussion.

Did you perform any sensitivity analysis to explore which model parameters, structures or compartments were most important for simulating spatial patterns and temporal dynamics?

3.1.2 First model adaptation (Models B – D) : Please describe what made you chose to make exactly these structural changes? Line 522: How can you argue that you significantly improve the spatial pattern of ET? Your ESP,ET might increase slightly from 0.18 to 0.23, but looking at the maps in Figure 11, Model F has the same pattern as Model A and none of them resemble the observed pattern.

Technical comments:

Figure 11 and similar figures: I suggest that you condense the figures to make less white space and thereby allow the reader to make a better visual examination of the observed and simulated patterns. You can skip the lat long degree for instance, they can be added to figure 1 instead.

Line 59: " to spatial pattern of" change to " to the spatial pattern of" or to " to spatial patterns of" Also in line 66 + 79 "spatial pattern and temporal dynamics" I suggest writing "spatial patterns"

Line 78: "for a large river systems" change to "for a large river system"

---

## Author Comment (AC2) · 14 Sep 2020

We thank the reviewer for his/her interest in our work and for the thoughtful and detailed feedback provided.

**Comment:**

The manuscript is very well written and nicely illustrated. It deals with an interesting topic of model structure and gaining information from satellite data in a data sparse basin.

**Reply:**

We highly appreciate this positive assessment of our manuscript. We will in the following address all specific comments in detail.

**Comment:**

It makes the paper less interesting that there are essentially no major changes to the simulated spatial patterns of ET across Model A-F (e.g. Figure 11) and that the general simulated spatial pattern does not resemble the observed in any way. It seems that you are not addressing the most important issues in your set of alternative models (B-F). The general interest of the manuscript would increase greatly if some of your hypothesis would at least produce a different pattern from Model A.

The simulated spatial pattern will be a reflection of both model structure and parametrizations scheme, in my experience mainly the latter. Therefore, I strongly suggest that you add a set of model setups that reflect different spatial parameterization schemes. In your discussion you address this limitation nicely, but I also feel that the manuscript would benefit greatly from an additional analysis illustrating the importance of model parameterization and parameter distribution on the simulated spatial patterns. Basically, even the most sophisticated model structure cannot be expected to reproduce a correct spatial pattern without a sound, flexible and spatially explicit parametrizations scheme.

I think you can logically add such an analysis to your manuscript in line with the idea of learning from satellite observations, by letting the observed spatial patterns guide your spatial parametrization approach. Such a parametrization scheme could also include transfer functions or simple spatial relations to known variables such as elevation, slope, soiltype, LAI etc.

**Reply:**

We completely agree with the observation of the Reviewer that there are indeed only limited improvements with respect to the spatial pattern of the evaporation when looking at Figure 11 in the manuscript. However, this figure includes only a selection of models with the best results applying the second calibration strategy using multiple variables. Additional figures were included in the Supplementary Material showing the spatial pattern for all Models A – F for both calibration strategies, hence using discharge (Figure S6 in the Supplementary Material) or multiple variables (Figure S10 in the Supplementary Material). With these graphs the effect of different model structures and calibration strategies on the spatial variability of the evaporation was illustrated.

On the one hand, when calibrating with respect to discharge only, the spatial pattern of the evaporation changed depending on the model structure, but remained poorly reproduced for all Models B - F

compared to the benchmark Model A (Figure S6 in the Supplementary Material). On the other hand, when calibrating with respect to multiple variables, the effect of the model structure was less significant (Figure S10 in the Supplementary Material). Only Model D showed clear differences compared to Model A, for example the dry season average evaporation was significantly overestimated in the wetland areas along the river in contrast to the observation (highlighted in Figure 1 here). In other words, the results in this study illustrate the spatial pattern of the evaporation did change with when changing the model structure or applying different calibration strategies.

However, the improvements remained very limited compared to the benchmark and the modelled spatial pattern remained poorly reproduced. We of course also completely agree with the reviewer, that at least some of the remaining problems are likely to be related to the actual distribution of parameters. As recommended by the Referee and mentioned in the discussion of our manuscript, this could, among others, be further improved by applying spatially distributed parameter sets. However, this will increase the number of calibration parameters considerably and hence also the degree of freedom such that many different combinations of parameters result in similar model performances, but do not necessarily reproduce all hydrological processes well. Therefore, it is important to have sufficient data to support spatially distributed parameters to avoid this problem of equifinality and improve the model realism. To limit the problems related to equifinalty, indeed a transfer function approach with global parameters, such as the MPR scheme developed for the mhm model (Samaniego et al., 2010;Kumar et al., 2013), could prove highly valuable. However, the design and choice of suitable and meaningful transfer functions in itself would require substantial additional analysis to assess the information content of different variables to support spatial parameter distribution (for example NDVI, LAI, topography, soil type, vegetation type or climate) and to test different distribution methods, which would warrant probably several standalone research papers. Therefore, as a first test, we analysed the effect of spatially distributing one calibration parameter related to the evaporation, the maximum interception storage ( $I_{max}$ ), using a linear transfer function with LAI data similar to previous studies (Samaniego et al., 2010) and using Model F as basis. While this influenced the spatial pattern of the evaporation, it did not improve significantly.

We will add an in-depth discussion on the risks and potentials of parameter distribution strategies in the revised manuscript. However, this paper focused on the added value of satellite-based evaporation and total water storage observations for model structure development and parameter selection. Therefore, additional analysis on parameter distribution strategies was considered outside the scope of this study.

Figure 1: Observed and modelled spatial variability of the normalised total evaporation averaged over all days within the dry season for Models A and D using the "optimal" parameter set with respect to multiple variables (*D*E,ESQcal).

**Comment:**

In sections 3.1.2 and 3.1.3 it is unclear why the different structural changes were applied. The title suggests that you are learning from satellite data, but it is not clear to me how you learn and how you used the satellite data to make new hypothesis about model structure. It is mentioned several times that you diagnose model deficiencies, however it is unclear to me how this is done. I believe this should be elaborated in a revised manuscript.

**Reply:**

In this study, the model structure was adapted iteratively based on the results of the benchmark Model A or subsequently developed models. Therefore, the paper was structured such that first the benchmark Model A was explained (Section 3.1.1), followed by a brief description of the model adaptations (Section 3.1.2 and 3.1.3). Based on the deficiencies of the benchmark Model A diagnosed and highlighted in Section 4.1.3, the first set of model adaptations were developed (Models B - D) as explained in Section 4.2. Similarly, based on the deficiencies diagnosed in Models B - D as explained in Section 4.3. Therefore, as the model adaptations were developed (Models E - F) as explained in Section 4.3. Therefore, as the model adaptations depended on the model results, they were explained only briefly in Section 3 and more detailed in Section 4.

However, we will update the manuscript to more specifically and explicitly emphasize the role of satellite observations when diagnosing deficiencies and changing the model structure. The satellite-based evaporation and total water storage observations were used for model evaluation with respect to their spatial and temporal variability to detect model deficiencies in these system-internal variables. In addition, satellite-based evaporation data was used to evaluate whether temporal variations in the evaporation from a specific hydrological response unit, in this case wetland dominated areas, were reproduced well. In all cases, the variables were normalised to focus on dynamic fluctuations and spatial pattern rather than absolute values to avoid incorporating bias uncertainties in the satellite data.

**Comment:**

An issue with the use of the SPAEF metric for the water storage anomaly might be, that the histogram component of the metric, might not be so meaningful when applied to the coarse spatial resolution of 1 deg., with very few grids. You could look into this by examining the three components of the metric separately. I do not suggest to put this analysis in the paper, but it might be mentioned in a discussion.

**Reply:**

Thank you for pointing this out. This could indeed provide some interesting insights. Upon closer inspection of  $E_{SP,E}$  and  $E_{SP,S}$  for the "optimal" parameter sets for Models A — F according to the first calibration strategy using discharge only, we discovered different ranges for the individual components as indicated in Table 1 here. According to these numbers, differences in  $E_{SP}$  were mainly a result of differences in  $\theta$  (coefficient of variation), whereas the component with the smallest difference was  $\alpha$  (Pearson correlation coefficient). The range in  $\gamma$  (fraction of histogram intersection) is indeed smaller for the total water storage where the grid size is larger compared to the evaporation. For future studies, it would be interesting to examine the different components more detailed to assess the overall information content of this model performance metric  $E_{SP}$  to identify feasible parameter sets across different spatial scales. We will elaborate on this in detail in the discussion.

Table 1: Overview of model performance ranges with respect to the spatial pattern ( $E_{SP}$ ) of evaporation and total water storage including the corresponding individual components ( $\alpha$ : Pearson correlation coefficient,  $\beta$ : coefficient of variation and  $\gamma$ : fraction of histogram intersection) using the "optimal" parameter sets according to the first calibration strategy using discharge only for Models A – F.

|     | Evaporation  | Total water storage |
|-----|--------------|---------------------|
| α   | 0.12 - 0.23  | 0.43 - 0.54         |
| 6   | 0.55 – 1.23  | 0.62 - 1.16         |
| Y   | 0.43 - 0.81  | 0.07 – 0.23         |
| Esp | -0.04 - 0.17 | -0.17 - 0.08        |

**Comment:**

Did you perform any sensitivity analysis to explore which model parameters, structures or compartments were most important for simulating spatial patterns and temporal dynamics?

**Reply:**

While we agree that an analysis of the respective sensitivities of individual aspects in the modelling processes could indeed provide additional insights into factors influencing the spatial and temporal variability we did not explicitly perform such a detailed analysis as this would have further inflated an already long, detailed and complex manuscript. However, the model results did shed light into the importance of several different aspects. For instance, when

considering the model parameter sensitivity, of the generated parameter sets, the best 5% were selected with respect to discharge or multiple variables. Other combinations of variables to identify feasible parameter sets, for example discharge and evaporation or only evaporation, were also tested but excluded from the manuscript as they did not add further value and to keep the story concise. Regardless of the calibration strategy, the modelled spatial pattern of the evaporation and total water storage remained significantly different from the observation when using the benchmark model. Also, the evaporation from wetland areas was reproduced poorly regardless of which variables were combined with discharge in the calibration procedure. This indicated these deficiencies were more likely a result of uncertainties in the model structure, parameterization or data rather than of the selected parameter sets. That is why the model structure was adjusted stepwise. While the spatial pattern mainly improved when incorporating multiple variables in the calibration procedure (compare Figures S6 and S10 in the Supplementary Material), the evaporation from wetland areas benefited the most from the changed model structure (Figure 10 in the manuscript).

A more systematic sensitivity analysis could provide valuable information on how to further improve the spatial and temporal variability of the system-internal variables, but this was considered outside the scope of this study. We will include this as recommendation in the revised manuscript.

**Comment:**

3.1.2 First model adaptation (Models B – D): Please describe what made you chose to make exactly these structural changes?

**Reply:**

The first set of model adaptations (Models B—D) depended on the results of the benchmark Model A. Therefore, the choice of adaptions was explained after having highlighted the deficiencies of Model A (Section 4.1.3) in Section 4.2. See also our reply on a previous comment on Sections 3.1.2 and 3.1.3. We acknowledge that this is not a conventional paper set-up, but we believe an iterative analysis warrants a partly iterative description of the steps.

**Comment:**

Line 522: How can you argue that you significantly improve the spatial pattern of ET? Your  $E_{SP,ET}$  might increase slightly from 0.18 to 0.23, but looking at the maps in Figure 11, Model F has the same pattern as Model A and none of them resemble the observed pattern.

**Reply:**

In this paper, the effect of using 1) multiple variables for model calibration and 2) alternative model structures on the spatial-temporal variability of among others evaporation was assessed. With respect to the spatial pattern, the results were illustrated with respect to the model performance values ( $E_{SP,E}$ ) and figures showing the spatial pattern. In the manuscript, only a

selection of these figures was shown (Figure 11), whereas in the Supplementary Material all remaining figures were included (Figures S6 and AS10).

In Line 522, we compared both calibration strategies for Model F. When calibrating Model F using discharge only, the spatial pattern of the evaporation was poorly reproduced (Figures 2b here and S6 in the Supplementary Material). This improved considerably when calibrating using multiple variables (Figures 2c here and 11 in the manuscript) as the evaporation was lower in the south-west and east of the basin similar to the observation. We will clarify this in the revised manuscript to avoid any confusion and also tone down the language to avoid confusion and misleading interpretation of our descriptions by the reader.

We absolutely agree with the Referee that the spatial pattern in the evaporation remain poorly reproduced. However, the goal of the statement mentioned by the Reviewer was to illustrate the added value of satellite observations to improve the representation of spatial and temporal variability of multiple variables. This paper showed that only limited improvements were observed in the spatial pattern with the chosen model structures and parameterization.

Figure 2: Spatial variability of the normalised total evaporation averaged over all days within the dry season according to WaPOR (observation) and Model F for both calibration strategies using discharge (Calibration strategy 1, *D*E,Ccal) or multiple variables (Calibration strategy 2, *D*E,ESQcal).

**Comment:**

Figure 11 and similar figures: I suggest that you condense the figures to make less white space and thereby allow the reader to make a better visual examination of the observed and simulated patterns. You can skip the lat long degree for instance, they can be added to figure 1 instead.

**Reply:**

Thank you for this feedback. We will condense these figures as much as possible to allow for a better visual comparison.

**Comment:**

Line 59: " to spatial pattern of" change to " to the spatial pattern of" or to " to spatial patterns of" Also in line 66 + 79 "spatial pattern and temporal dynamics" I suggest writing "spatial patterns"

**Reply:**

Thank you for pointing this out. We will correct this in the revised manuscript.

**Comment:**

Line 78: "for a large river systems" change to "for a large river system"

**Reply:**

Thank you for pointing this out. We will correct this in the revised manuscript.

**Literature**

Kumar, R., Samaniego, L., and Attinger, S.: Implications of distributed hydrologic model parameterization on water fluxes at multiple scales and locations, Water Resources Research, 49, 360-379, 10.1029/2012WR012195, 2013.

Samaniego, L., Kumar, R., and Attinger, S.: Multiscale parameter regionalization of a grid-based hydrologic model at the mesoscale, Water Resources Research, 46, 10.1029/2008WR007327, 2010.

---

## Author Response (AR2)

We thank the Editor and Reviewer for their feedback to further improve the paper. The paper was modified taking the comments into account as shown in the marked-up version of the modified paper.

**Responses to the Editor**

*Comment:*

*Dear authors,*
*as mentioned by the reviewer and also having a look myself, I think you did a good job in addressing most of the points the reviewers mentioned in the first round of the reviews. I can also see your point, that some of the suggested changes are not possible without redoing most of the analysis. Hence, I think the paper could be accepted if - as request from the reviewer and supported by myself, the conclusion will be adapted. As mentioned my the reviewer, it would be appropriate to highlight, that due to the limitations of the selected study area, model type and data available, the study will mainly be a novel exemplification of a framework for exploiting satellite data for improved process representation - but certainly cannot be generalized. Meaning that the main interest of this paper is on the idea and framework, and less on the actual improved process understanding. The model, data and approach cannot claim to have increased the process understanding significantly. However, you have exemplified a very interesting approach to potentially use satellite information for increased process understanding. Please make the requested changes and highlight the changes you made in the paper.*

Response:

Thank you for this positive assessment. We agree, as a result of the limited model hypotheses tested in this study, Model F performed best among the hypotheses tested here but in future studies an alternative hypothesis may perform even better. However, this study showed the added value of satellite observations for stepwise model development which contributes to improved process understanding. We have modified the conclusion in the revised version of the paper to address this.

**Responses to Anonymous Reviewer #1**

*Comment*:

*The authors have responded in detail to the questions raised and fundamentally maintained the structure of the paper and argued that some of the suggestions for major revisions are outside the scope of the paper.*
*I appreciate the added discussion on limitations of the study and in particular the data scarce and uncertain study area. I especially appreciate the added analysis of alternative precipitation data (figure S15) and an attempt to address parameter distribution (Model G, Figure S14). I would still argue that the paper would have contributed more towards actually learning from satellite data, if the parametrization had been addressed, both through sensitivity analysis towards different performance metrics and different parametrisation schemes. This would, as argued by the authors require an undesired expansion of the manuscript, and would consequently require a restructuring of the entire study and reduction of the already presented analysis.*
*I find the manuscript ready for publication, but I do have one request, which I find reasonable. That is to address the limitations of the study in the conclusions. It would be appropriate to highlight, that due to the limitations of the selected study area, model type and data available, the study will mainly be a novel exemplification of a framework for exploiting satellite data for improved process representation. Meaning that the main interest of this paper is on the idea and framework, and much less on the actual*

*improved process understanding. I simply do not believe the model, data and approach can clearly claim to have increased the process understanding significantly. However, you have exemplified a very interesting approach to potentially use satellite information for increased process understanding. E.g.Your Model F performs best, but weather groundwater upwelling is the correct process, is not exhaustively examined, especially when considering all the uncertainties.*

Response:

Thank you for this positive assessment. In this paper we tested a limited number of model hypotheses to keep the paper as concise as possible. Based on our results, Model F is the best performing model among the hypotheses tested in this study. However, when considering alternative hypotheses in future studies, this model may be rejected in favour of a new model hypothesis. Therefore, we agree a more comprehensive analysis is needed to further improve the representation of groundwater upwelling and hence our understanding of the hydrological process. Nevertheless, this study showed that satellite observations provide valuable information not only for model calibration, but also stepwise model development which contributes to improved process understanding. We have modified the conclusion in the revised version of the paper to address this.